# *Candida albicans* stimulates formation of a multi-receptor complex that mediates epithelial cell invasion during oropharyngeal infection

Quynh T. Phan[1], Norma V. Solis[1], Max V. Cravener[2], Marc Swidergall[1,3], Jianfeng Lin[1], Manning Y. Huang[4], Hong Liu[1], Shakti Singh[1], Ashraf S. Ibrahim[1,3], Massimiliano Mazzone[5,6], Aaron P. Mitchell[2], Scott G. Filler [1,3] *

1 Institute for Infection and Immunity, Lundquist Institute for Biomedical Innovation at Harbor-UCLA Medical Center, Torrance, California, United States of America, 2 Department of Microbiology, University of Georgia, Athens, Georgia United States of America, 3 Department of Medicine, David Geffen School of Medicine at UCLA, Los Angeles, California, United States of America, 4 Department of Biochemistry and Biophysics, University of California, San Francisco, California, United States of America, 5 Laboratory of Tumor Inflammation and Angiogenesis, Center for Cancer Biology, VIB, Leuven, Belgium, 6 Laboratory of Tumor Inflammation and Angiogenesis, Center for Cancer Biology, Department of Oncology, KU Leuven, Leuven, Belgium

* sfiller@ucla.edu

**Data Availability Statement:** The authors confirm that all data underlying the findings are fully available without restriction. All relevant data are

## Abstract

Fungal invasion of the oral epithelium is central to the pathogenesis of oropharyngeal candidiasis (OPC). *Candida albicans* invades the oral epithelium by receptor-induced endocytosis but this process is incompletely understood. We found that *C. albicans* infection of oral epithelial cells induces c-Met to form a multi-protein complex with E-cadherin and the epidermal growth factor receptor (EGFR). E-cadherin is necessary for *C. albicans* to activate both c-Met and EGFR and to induce the endocytosis of *C. albicans*. Proteomics analysis revealed that c-Met interacts with *C. albicans* Hyr1, Als3 and Ssa1. Both Hyr1 and Als3 are required for *C. albicans* to stimulate c-Met and EGFR in oral epithelial cells in vitro and for full virulence during OPC in mice. Treating mice with small molecule inhibitors of c-Met and EGFR ameliorates OPC, demonstrating the potential therapeutic efficacy of blocking these host receptors for *C. albicans*.

## Author summary

During oropharyngeal candidiasis (OPC), *Candida albicans* expresses the Als3 invasin, which interacts with multiple oral epithelial cell receptors including E-cadherin, the epidermal growth factor receptor (EGFR) and HER2. Activation of these receptors leads to two oral epithelial cell responses, the endocytosis of *C. albicans* hyphae and production of pro-inflammatory cytokines and chemokines. By analyzing the transcriptome of oral epithelial cells infected with *C. albicans*, we saw evidence that c-Met, the receptor for hepatocyte growth factor, was activated, leading us to investigate whether c-Met functions as a

within the paper and its Supporting Information files.

**Funding:** This work was supported in part by the National Institute of Dental and Craniofacial Research, grant R01DE026600 to SGF and APM and grant R01DE031382 to MS from the National Institutes of Health, USA. The funders had no role in study design, data collection and analysis, decision to publish, or preparation of the manuscript.

**Competing interests:** The authors have declared that no competing interests exist.

receptor for *C. albicans*. We found that in response to candidal infection, c-Met and EGFR form a multi-protein complex with E-cadherin. The presence of E-cadherin in this complex is required for *C. albicans* to activate c-Met and EGFR and to induce endocytosis of the organism. Activation of c-Met and EGFR requires not only Als3, but also a second *C. albicans* surface protein, Hyr1. A mutant lacking Hyr1 has reduced virulence in the mouse model of OPC and combined pharmacologic blockade of EGFR and c-Met reduces the severity of OPC. Thus c-Met functions as an epithelial cell receptor for *C. albicans* that interacts with both Als3 and Hyr1.

## Introduction

Oropharyngeal candidiasis (OPC), caused predominantly by *Candida albicans*, afflicts patients with HIV/AIDS, Sjogren's syndrome, inhaled corticosteroid use, and cancer of the head and neck [1–4]. Azole antifungal drugs are currently the mainstay of therapy for patients with OPC [5]. However, the emergence of azole resistance [6–8] necessitates new strategies to prevent and treat this disease.

A key step in the pathogenesis of OPC is fungal invasion of the epithelial cell lining of the oropharynx [9,10]. This process occurs both when a new focus of infection is initiated and as an established fungal lesion enlarges. *C. albicans* can invade epithelial cells by two mechanisms. One is active penetration, in which a progressively elongating hypha pushes its way into the host cell [11,12]. The other is receptor-induced endocytosis, in which *C. albicans* binds to receptors on the surface of the epithelial cell, triggering the rearrangement of epithelial cell microfilaments and inducing the formation of pseudopods that surround the organism and pull it into the epithelial cell [13–15]. Multiple epithelial cell receptors for *C. albicans* have been identified including the ephrin type-A receptor 2 (EphA2), E-cadherin, HER2, and the epidermal growth factor receptor (EGFR). EphA2 binds to β-glucans, while E-cadherin, HER2, and EGFR interact with *C. albicans* Als3 and Ssa1 [16–19]. All of these receptors appear to function in the same pathway to mediate the endocytosis of *C. albicans* [17,18].

While analyzing the transcriptional response of oral epithelial cells to *C. albicans* infection, we discovered that the hepatocyte growth factor (HGF) pathway was activated [20]. HGF is the ligand for the c-Met receptor tyrosine kinase. This finding prompted us to investigate the role of c-Met in the endocytosis of *C. albicans*. Here, we report that c-Met functions as a host cell receptor for *C. albicans*. It interacts not only with *C. albicans* Als3, but also with Hyr1, which we determined is required for maximal epithelial cell invasion. In response to *C. albicans*, c-Met forms a multi-protein complex with EGFR and E-cadherin that is required for the endocytosis of the organism in vitro and for maximal virulence during OPC.

## Results

### c-Met functions in conjunction with EGFR to induce endocytosis of *C. albicans*

To ascertain whether c-Met functions as an epithelial cell receptor for *C. albicans*, we infected the immortalized OKF6/TERT-2 oral epithelial cell line [21] with *C. albicans* hyphae and then used indirect immunofluorescence to detect the localization of c-Met. We observed that c-Met accumulated around the hyphae in a similar fashion to EGFR and E-cadherin (Figs 1A and S1A). By immunoblotting of oral epithelial cell lysates with phosphospecific antibodies, we determined that *C. albicans* infection induced c-Met autophosphorylation that was detectable

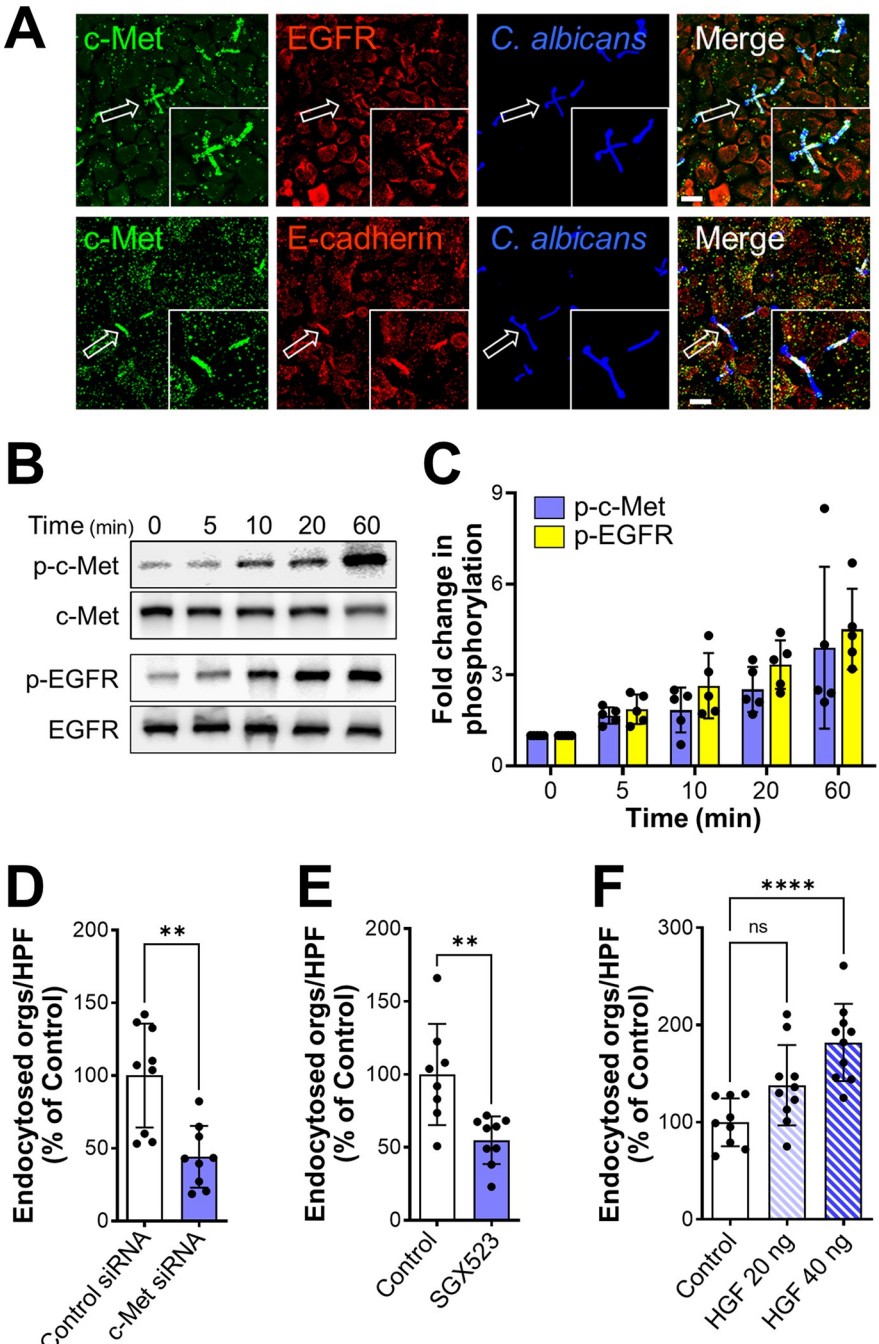

**Fig 1. *C. albicans* activates c-Met in oral epithelial cells.** (A) Confocal microscopic images of OKF6/TERT-2 oral epithelial cells infected with *C. albicans* SC5314 for 20 min. c-Met, the epidermal growth factor receptor (EGFR), and E-cadherin were detected by indirect immunofluorescence using specific antibodies. Arrows point to the organisms in the magnified insets. Scale bar 10 μm. (B and C) Immunoblot analysis showing the time course of the phosphorylation of c-Met and EGFR in oral epithelial cells induced by *C. albicans* germ tubes (B). Densitometric analysis of 4 immunoblots (C). Data are mean ± SD. (D and E) Knockdown of c-Met with siRNA (D) or inhibition of c-Met signaling with SG523 (E) in oral epithelial cells inhibits the endocytosis of *C. albicans*. (F) Stimulation of oral epithelial cells with recombinant hepatocyte growth factor (HGF) enhances the endocytosis of *C. albicans*. Data in (D-F) are the mean ± SD of three experiments, each performed in triplicate. $**p < 0.01$, $****p < 0.0001$, ns; not significant (two-tailed Student's t test [D and E] or one-way ANOVA with Sidak's multiple comparisons test [F]).

within 5 min and persisted for at least 60 min (Fig 1B and 1C). The time course of *C. albicans*-induced phosphorylation of c-Met paralleled that of the phosphorylation of EGFR. Collectively, these results indicate that contact with *C. albicans* hyphae activates c-Met similarly to EGFR.

The functional significance of c-Met activation was analyzed by assessing its effect on the endocytosis of *C. albicans*. Both siRNA knockdown of c-Met and blockade of c-Met kinase activity with the specific tyrosine kinase inhibitor SGX523 decreased the endocytosis of *C. albicans* by approximately 50% (Fig 1D and 1E). These interventions had no effect on the number of cell-associated organisms, a measure of adherence (S1B and S1C Fig). Of note, knockdown of c-Met did not affect cellular levels of EGFR or E-cadherin (S1D Fig). Activating c-Met even further by adding its natural ligand HGF at the same time as the *C. albicans* enhanced endocytosis in a dose-dependent manner, independently of adherence (Figs 1F, S1E and S1F). Therefore, c-Met activity partially governs the endocytosis of *C. albicans*.

To analyze the relationship between c-Met and EGFR, we investigated the effects of different inhibitors on *C. albicans*-induced phosphorylation of these receptors. When the epithelial cells were treated with gefitinib, a specific inhibitor of EGFR kinase activity, and then infected with *C. albicans*, EGFR phosphorylation was reduced to basal levels but c-Met phosphorylation remained unchanged (Fig 2A and 2B). When the epithelial cells were treated with SGX523, c-Met phosphorylation was inhibited but EGFR phosphorylation was not (Fig 2A and 2C). Thus, *C. albicans* induces the phosphorylation of c-Met independently of EGFR.

Next, we investigated the combined effects of these inhibitors on the endocytosis of *C. albicans* to determine the functional relationship between EGFR and c-Met. Incubating epithelial cells with either SGX523 or gefitinib alone reduced the endocytosis of *C. albicans* by a similar amount (Fig 2D). Incubating the cells with SGX523 and gefitinib in combination had an additive effect, inhibiting endocytosis significantly more than SGX253 alone. This reduction in epithelial cell uptake was due solely to inhibition of receptor-induced endocytosis because the combination of SGX523 and gefitinib had no effect on the invasion of *C. albicans* into fixed epithelial cells, which occurs by the process of active penetration (S2A and S2B Fig). The lack of effect of these inhibitors on active penetration also indicates that they had no effect on *C. albicans* germination and hyphal elongation. Also, neither inhibitor consistently reduced *C. albicans* adherence to the epithelial cells, although the combination of SGX523 and gefitinib modestly decreased *C. albicans* adherence in some experiments (S2B and S2C Fig). The additive inhibitory effect of gefitinib and SGX523 on the endocytosis of *C. albicans* is consistent with the model that both c-Met and EGFR are required for maximal uptake of this organism.

To further investigate the functional interactions between c-Met and EGFR, we used the NIH/3T3 mouse fibroblastoid cell line. When these cells were transfected with human c-Met, they endocytosed *C. albicans* similarly to control cells that had been transfected with GFP (Fig 2E). By contrast, when NIH/3T3 cells that expressed human EGFR and HER2 were transfected with c-Met, they endocytosed significantly more organisms than the control cells (Fig 2F). Transfection of either cell line with c-Met had minimal effects on adherence (S2D and S2E Fig). Collectively, these results indicate that the presence of EGFR is necessary for c-Met to induce endocytosis.

Using indirect immunofluorescence with an anti-phospho-c-Met antibody, we analyzed how the presence of EGFR and HER2 affected *C. albicans*-induced phosphorylation of c-Met in NIH/3T3 cells. As expected, no c-Met phosphorylation was detected in control NIH/3T3 cells that expressed either no human receptors or human EGFR and HER2 (Fig 2G). While *C. albicans* infection induced modest c-Met phosphorylation in NIH/3T3 cells that expressed human c-Met alone, it induced strong c-Met phosphorylation in cells that expressed human c-Met, EGFR, and HER2. Thus, the presence of EGFR and HER2 is required for *C. albicans* to induce maximal phosphorylation of c-Met.

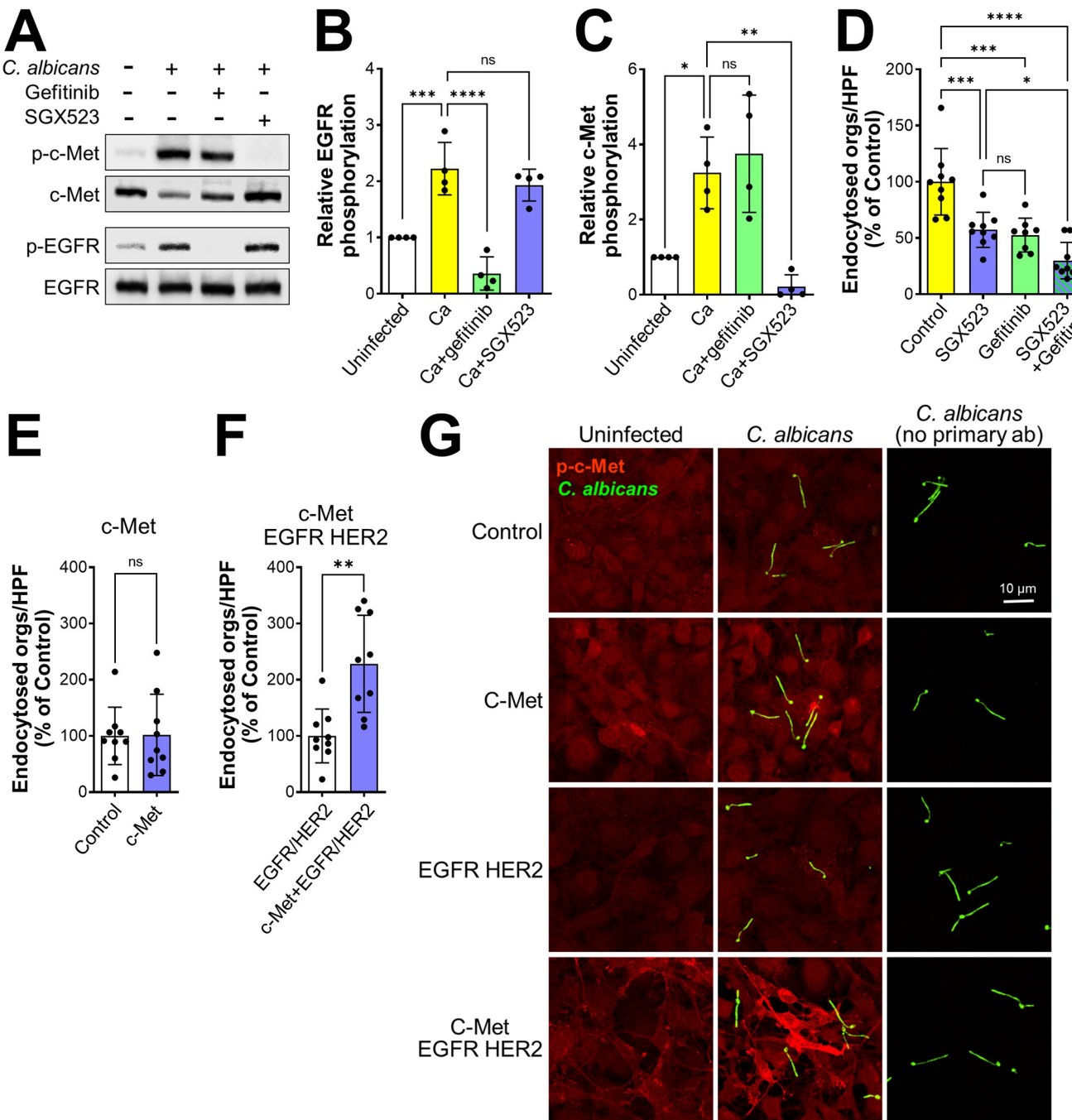

**Fig 2. Functional interactions among c-Met, EGFR, and E-cadherin during the endocytosis of *C. albicans*.** (A-C) Immunoblot analysis showing the effects of the EGFR inhibitor gefitinib and the c-Met inhibitor SGX523 on *C. albicans*-induced phosphorylation of c-Met and EGFR in oral epithelial cells after 20 min of infection. Representative immunoblot (A), densitometric analysis of 4 immunoblots showing the phosphorylation of EGFR (B) and c-Met (C). Data are mean ± SD. (D) Effects of SGX523 and gefitinib on the endocytosis of *C. albicans* by oral epithelial cells. (E and F) Endocytosis of *C. albicans* by NIH/3T3 cells that expressed human c-Met (E) or human c-Met, EGFR, and HER2 (F). (G) Confocal micrographs of NIH3T3 cells expressing no human receptors (control), human c-Met, human EGFR and HER2, or human c-Met, EGFR, and HER2. The cells were infected with wild-type *C. albicans* germ tubes for 20 min, after which phosphorylated c-Met was detected with a phosphospecific anti-c-Met antibody. Scale bar 10 μm. Results in (D-F) are the mean ± SD of three experiments, each performed in triplicate. *$p < 05$, **$p < 0.01$, ***$p < 0.001$, ****$p < 0.0001$, ns; not significant (one-way ANOVA with Sidak's multiple comparisons test [B-D] or two-tailed Student's t test [E and F]).

## E-cadherin interacts with both c-Met and EGFR

Our previous finding that E-cadherin functions in the same pathway as EGFR to mediate the endocytosis of *C. albicans* [17] prompted us to evaluate the role of E-cadherin in *C. albicans*-induced phosphorylation of c-Met and EGFR. When epithelial cell E-cadherin was knocked down with siRNA prior to *C. albicans* infection, there was reduced phosphorylation of both c-Met and EGFR (Fig 3A–3C). While E-cadherin knockdown also decreased the endocytosis of *C. albicans*, the combination of E-cadherin knockdown with either SGX523 or gefitinib did not inhibit endocytosis further (Fig 3D and 3E), and there was no effect on adherence (S2F and S2G Fig). Collectively, these results support the model that E-cadherin is a common member of the c-Met and EGFR signaling pathways that mediate the endocytosis of *C. albicans*.

These results suggest that c-Met, EGFR, and E-cadherin might be components of a multi-protein complex. To test this hypothesis, we used a proximity ligation assay, which forms a fluorescent spot when two proteins are within 40 nm of each other [22,23]. In uninfected epithelial cells, there was low level association of c-Met, EGFR, and E-cadherin that appeared to be randomly distributed throughout the cells (Fig 4A and 4B). When the epithelial cells were infected with *C. albicans*, there was a marked increase in the association of E-cadherin with both c-Met and EGFR, and a modest but still significant increase in the association of c-Met with EGFR. Interestingly, these protein complexes accumulated both in the vicinity of *C. albicans* hyphae and in cells that did not appear to be in contact with the organism. No signals were observed the proximity ligation assay was performed using control mouse and rabbit IgG (S3A Fig).

To verify the association of E-cadherin with c-Met and EGFR, we performed coimmunoprecipitation experiments on lysates of oral epithelial cells. When lysates of uninfected epithelial cells were immunoprecipitated with an anti-c-Met antibody, low levels of both EGFR and E-cadherin were coimmunoprecipitated (Fig 4C–4E). When the epithelial cells were infected with *C. albicans*, there was increased coimmunoprecipitation of both EGFR and E-cadherin with c-Met. This coimmunoprecipitation was significantly reduced in epithelial cells in which E-cadherin was depleted with siRNA. We obtained similar results in reciprocal experiments in which c-Met and E-cadherin were coimmunoprecipitated using an anti-EGFR antibody (S3B–S3D Fig). Taken together, these results indicate that *C. albicans* infection enhances the formation of a multi-protein complex that contains c-Met, EGFR, and E-cadherin. They also suggest that E-cadherin is a central component of this complex and is required for *C. albicans* to activate both c-Met and EGFR.

## Hyr1 is a *C. albicans* ligand for c-Met

We set out to identify the *C. albicans* ligands for c-Met. The phosphorylation of c-Met and EGFR was induced by *C. albicans* hyphae, but not by yeast-phase organisms, indicating that the *C. albicans* ligand for c-Met must be expressed predominantly by hyphae (Fig 5A–5C). Because Als3 and Ssa1 are surface expressed only on hyphae and are required for *C. albicans* to activate EGFR in oral epithelial cells [17], we tested a *C. albicans* als3Δ/Δ ssa1Δ/Δ mutant and found that it induced minimal phosphorylation of both EGFR and c-Met (Fig 5D–5F), suggesting that Als3 and/or Ssa1 are required for the activation of both receptors.

Next, we considered the possibility that there may be an additional fungal ligand for c-Met. To identify this ligand, we used a far Western blotting approach in which cell wall proteins were isolated from wild-type and als3Δ/Δ ssa1Δ/Δ mutant strains, separated by non-reducing SDS-PAGE, and transferred to nylon membranes that were probed with recombinant c-Met. We found that c-Met bound to a single broad band in lanes containing cell wall proteins from hyphae of the wild-type strain or the als3Δ/Δ ssa1Δ/Δ mutant, but not wild-type yeast (Fig 5G).

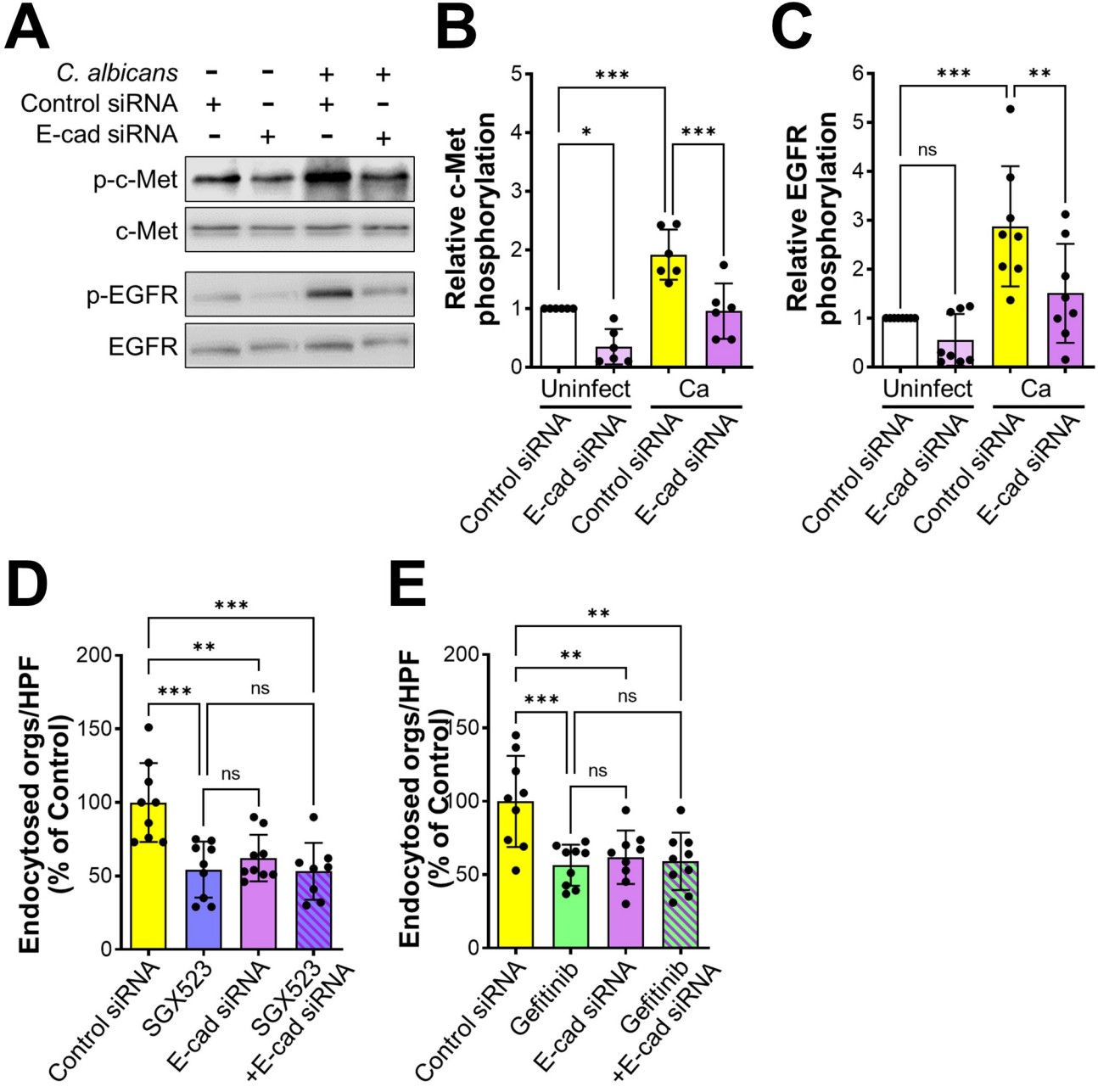

**Fig 3. E-cadherin is required for *C. albicans* to activate c-Met and EGFR during the endocytosis of *C. albicans*.** (A-C) Knockdown of E-cadherin by siRNA inhibits the phosphorylation of c-Met and EGFR in oral epithelial cells infected with *C. albicans*. Representative immunoblot (A). Densitometric analysis of 5 immunoblots showing the phosphorylation of c-Met (B) and EGFR (C). Results are mean ± SD. (D and E) Effects of inhibiting c-Met (D) and EGFR (E) in combination with siRNA knockdown of E-cadherin on the endocytosis of *C. albicans* by oral epithelial cells. Results in (D and E) are the mean ± SD of three experiments, each performed in triplicate. *$p < 05$, **$p < 0.01$, ***$p < 0.001$, ns; not significant (one-way ANOVA with Sidak's multiple comparisons test.

Using tandem mass spectrometry, we determined the identities of the proteins in the bands from wild-type and *als3*Δ/Δ *ssa1*Δ/Δ hyphae. As expected, both bands contained multiple proteins (S1 Table), so we focused on proteins that are known to be expressed on the fungal cell surface. Both Als3 and Ssa1 were detected in the band containing proteins from the wild-type strain, but not the *als3*Δ/Δ *ssa1*Δ/Δ mutant, thus validating the assay (Table 1). We also found

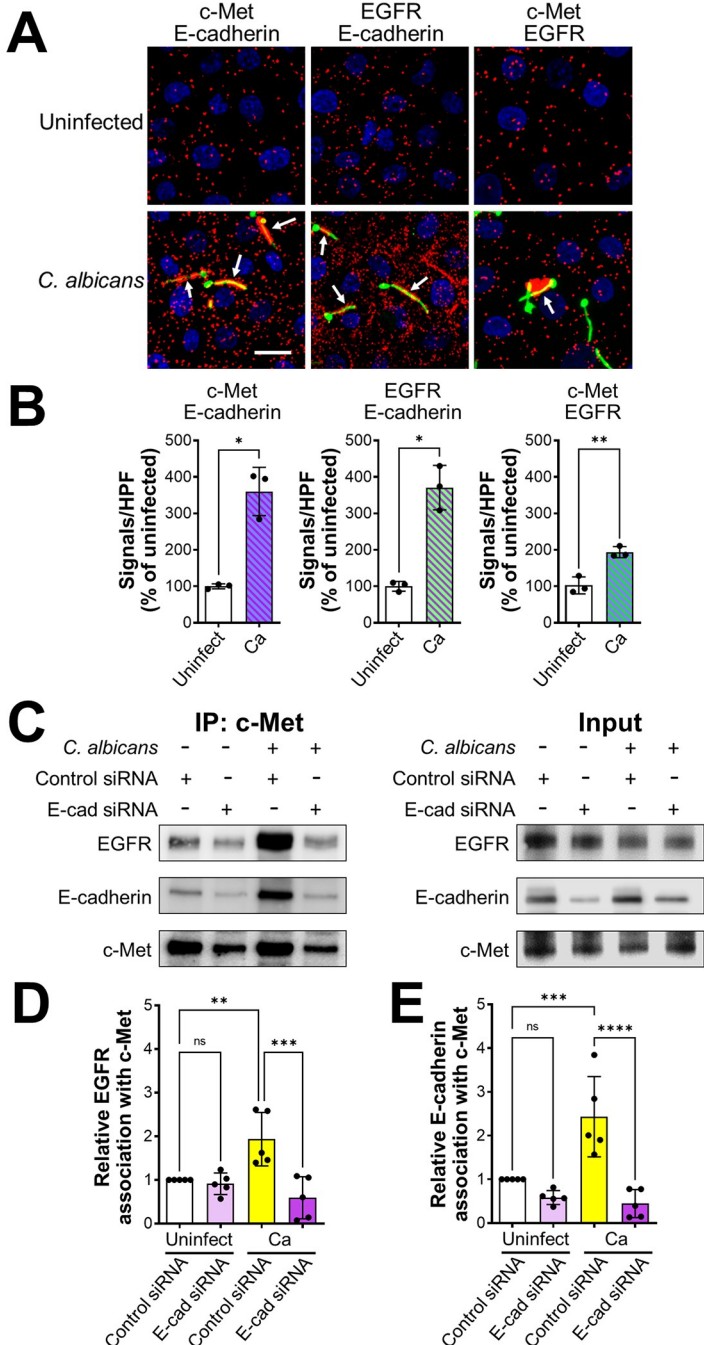

**Fig 4. *C. albicans* induces c-Met, EGFR, and E-cadherin to form a multiprotein complex.** (A-C) Proximity ligation assay showing the interaction of c-Met with E-cadherin, EGFR with E-cadherin, and c-Met with EGFR in oral epithelial cells with and without 20-min infection with *C. albicans*. Confocal microscopic images (A). Scale bar 10 μm. Signal counts (B). Proximity ligation assay showing the interaction of c-Met with E-cadherin, EGFR with E-cadherin, and c-Met with EGFR in oral epithelial cells with and without 20-min infection with *C. albicans*. (A) Confocal microscopic images. Scale bar 10 μm. (B) Signal counts. (C-E) Co-immunoprecipitation experiments in oral epithelial cells transfected with control or E-cadherin (E-cad) siRNA and then infected with *C. albicans* for 20 min. (C) Representative immunoblots of proteins obtained by immunoprecipitation with an anti-c-Met antibody. (D and E) Densitometric analysis of 5 immunoblots. Results are mean ± SD. **$p < 0.01$, ***$p < 0.001$, ****$p < 0.0001$, ns; not significant (two-tailed Student's t test [B] or one-way ANOVA with Sidak's multiple comparisons test [D and E]).

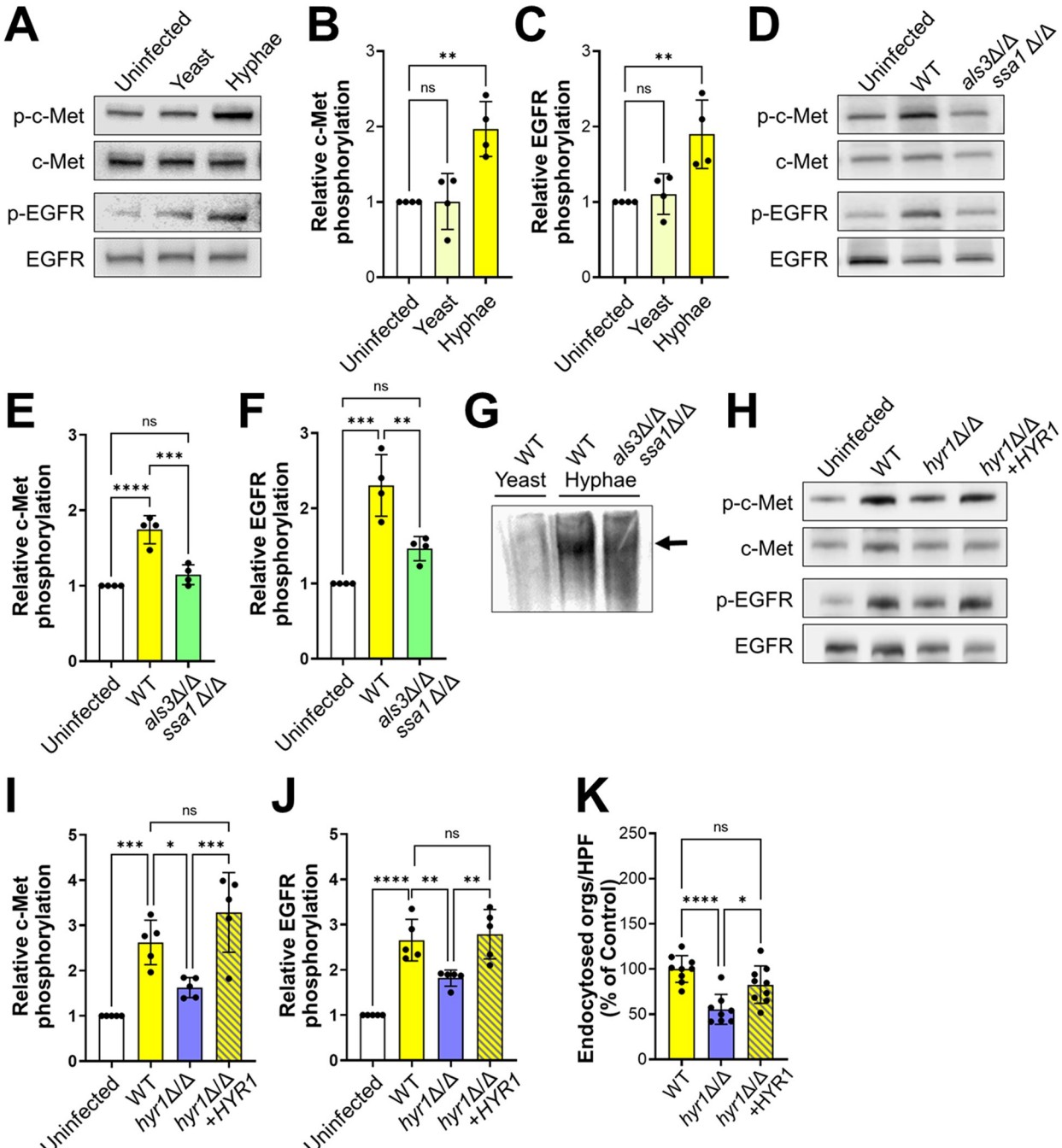

**Fig 5. Hyr1 interacts with c-Met.** (A-C). *C. albicans* germ tubes stimulate phosphorylation of c-Met and EGFR after 20 min of infection. Representative immunoblots (A). Densitometric analysis of 4 immunoblots showing the phosphorylation of c-Met (B) and EGFR (C) induced by *C. albicans* yeast and hyphae. (D-F) An *als3 Δ/Δ ssa1Δ/Δ* mutant does not induce phosphorylation of c-Met and EGFR. Representative immunoblots (D). Densitometric analysis of 4 immunoblots showing the phosphorylation of c-Met (E) and EGFR (F) induced by the indicated strains of *C. albicans*. (G) Far Western blot showing proteins from the indicated *C. albicans* morphotypes and strains that were recognized by recombinant c-Met. Arrow indicates the protein band. (H-J) Hyr1 is required for maximal phosphorylation of c-Met and EGFR. Representative immunoblots (H). Densitometric analysis of 4 immunoblots showing the phosphorylation of c-Met (I) and EGFR (J) induced by the indicated strains of *C. albicans*. (K) Hyr1 is required for maximal endocytosis of *C. albicans*. Results in (B, C, E, F, I-K) are mean ± SD. WT, wild type; *p < 0.05, **p < 0.01, ***p < 0.001, ****p, 0.0001, ns; not significant (one-way ANOVA with Sidak's multiple comparisons test).

**Table 1. Selected *C. albicans* surface proteins identified in the far Western blotting experiment.**

| Protein | Description | No. of Peptides Wild-type | No. of Peptides *als3Δ/Δ ssa1Δ/Δ* |
|---|---|---|---|
| Hyr1 | GPI-anchored hyphal cell wall protein | 8 | 6 |
| Ssa1 | HSP 70 family chaperone, role in host cell entry | 8 | 0 |
| Plb3 | GPI-anchored cell surface phospholipase B | 4 | 4 |
| Eno1 | Enolase, major cell surface antigen | 4 | 3 |
| Pra1 | Cell surface protein that sequesters zinc from host tissue | 4 | 0 |
| Pga31 | GPI anchored cell wall protein | 3 | 3 |
| Als1 | Agglutinin-like protein 1, cell surface adhesin | 2 | 3 |
| Als3 | Agglutinin-like protein 3, cell surface adhesin/invasin | 2 | 0 |
| Phr1 | Cell-surface glycosidase | 0 | 2 |

Als1 in the bands from both *C. albicans* strains. Notably, multiple Hyr1 peptides were identified in the bands from both strains. This hyphal-specific protein has previously been found to inhibit *C. albicans* killing by neutrophils and contribute to virulence in mouse models of both disseminated candidiasis and OPC, but its host cell receptor was unknown [24,25].

We focused our subsequent investigations on Hyr1 and Als3. Functional analysis supported the hypothesis that Hyr1 is a c-Met ligand. An *hyr1Δ/Δ* deletion mutant induced minimal phosphorylation of c-Met (Fig 5H and 5I). This mutant also had reduced capacity to stimulate the phosphorylation of EGFR as compared to the wild-type strain (Fig 5H and 5J). Although the *hyr1Δ/Δ* mutant had normal adherence, it was endocytosed poorly by oral epithelial cells, (Figs 5K and S4A). The phosphorylation and endocytosis defects of the *hyr1Δ/Δ* mutant were restored by complementing the mutant with a wild-type copy of *HYR1*. Collectively, these data support the model that *C. albicans* Als3 and Hyr1 cause c-Met, EGFR, and E-cadherin to form a multi-component complex that is essential for *C. albicans*-induced activation of c-Met and EGFR and the subsequent epithelial cell endocytosis of the fungus (Fig 6).

## Hyr1-mediated invasion is dispensable for epithelial cell damage

Both Als3 and the candidalysin pore forming toxin are required for *C. albicans* to induce maximal phosphorylation of EGFR in oral epithelial cells and to damage these cells [17,22,26,27]. We investigated whether candidalysin is also required for *C. albicans* to induce phosphorylation of c-Met by testing an *ece1Δ/Δ* mutant that does not make this toxin. Although the *ece1Δ/Δ* mutant induced minimal phosphorylation of EGFR, it stimulated the phosphorylation of c-Met similarly to the wild-type strain, indicating that candidalysin is dispensable for c-Met activation (S4B–S4D Fig).

By mutant analysis, we investigated the functional relationship between Hyr1 and Als3 in activating and damaging oral epithelial cells. Both the *als3Δ/Δ* single mutant and the *hyr1Δ/Δ als3Δ/Δ* double mutant induced minimal phosphorylation of c-Met and EGFR, similarly to the *hyr1Δ/Δ* single mutant (Fig 7A–7C). While both the *als3Δ/Δ* and *hyr1Δ/Δ* single mutants were endocytosed poorly by the oral epithelial cells, the invasion defect of the *als3Δ/Δ* mutant was greater than that of the *hyr1Δ/Δ* mutant (Fig 7D). The *als3Δ/Δ* mutant was endocytosed so poorly that deletion of *HYR1* in this mutant did not further decrease endocytosis. Similar results were seen with the adherence of these strains (S4E Fig). Thus, we were unable to determine if Als3 and Hyr1 make additive contributions to epithelial cell invasion and adherence in vitro.

It has been postulated that Als3 is required for epithelial cell damage because it is necessary for *C. albicans* to invade the host cell, leading to the formation of an invasion pocket into

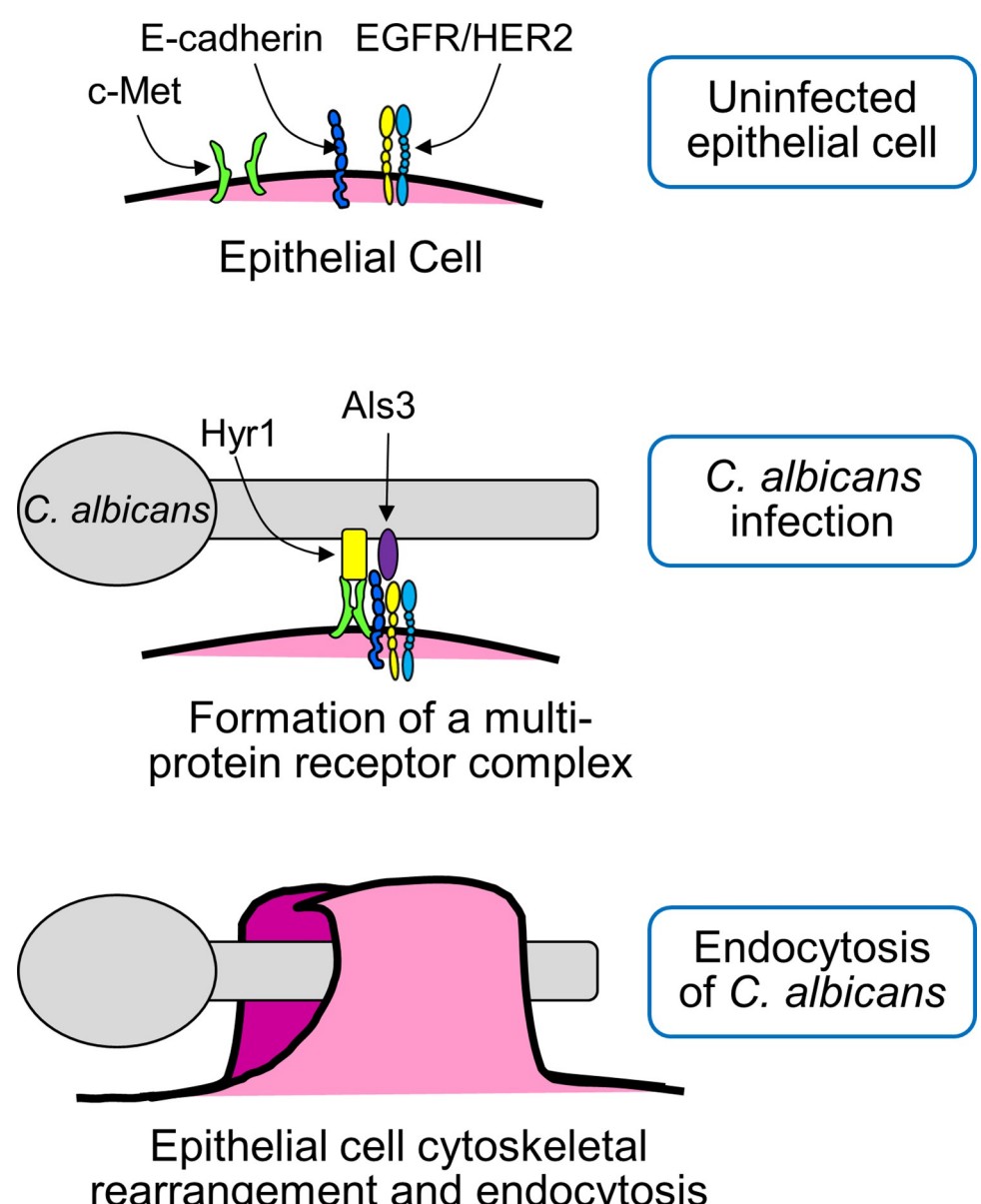

**Fig 6. Model of how *C. albicans* induces its own endocytosis by oral epithelial cells.** In uninfected epithelial cells, c-Met, E-cadherin, and EGFR/HER2 are not associated with each other in the cell membrane. When a cell is infected with *C. albicans*, Hyr1 and Als3 on the fungal cell surface cause c-Met, E-cadherin, and EGFR/HER2 to form a multi-protein complex. This complex contains activated (phosphorylated) c-Met and EGFR, which induce epithelial cell cytoskeletal rearrangement and the subsequent endocytosis of the fungus.

which candidalysin can accumulate to a sufficiently high concentration to form pores in the plasma membrane [22,28]. Because Hyr1 is required for normal epithelial cell invasion, we compared the capacity of the *hyr1Δ/Δ*, *als3Δ/Δ*, and *hyr1Δ/Δ als3Δ/Δ* mutants to damage oral epithelial cells in vitro. Although the *als3Δ/Δ* single mutant and the *hyr1Δ/Δ als3Δ/Δ* double mutant caused virtually no epithelial cell damage, the *hyr1Δ/Δ* single mutant induced the same amount of damage as the wild-type strain (Fig 7E). These results suggest that invasion mediated by Hyr1 is dispensable for inducing epithelial cell damage.

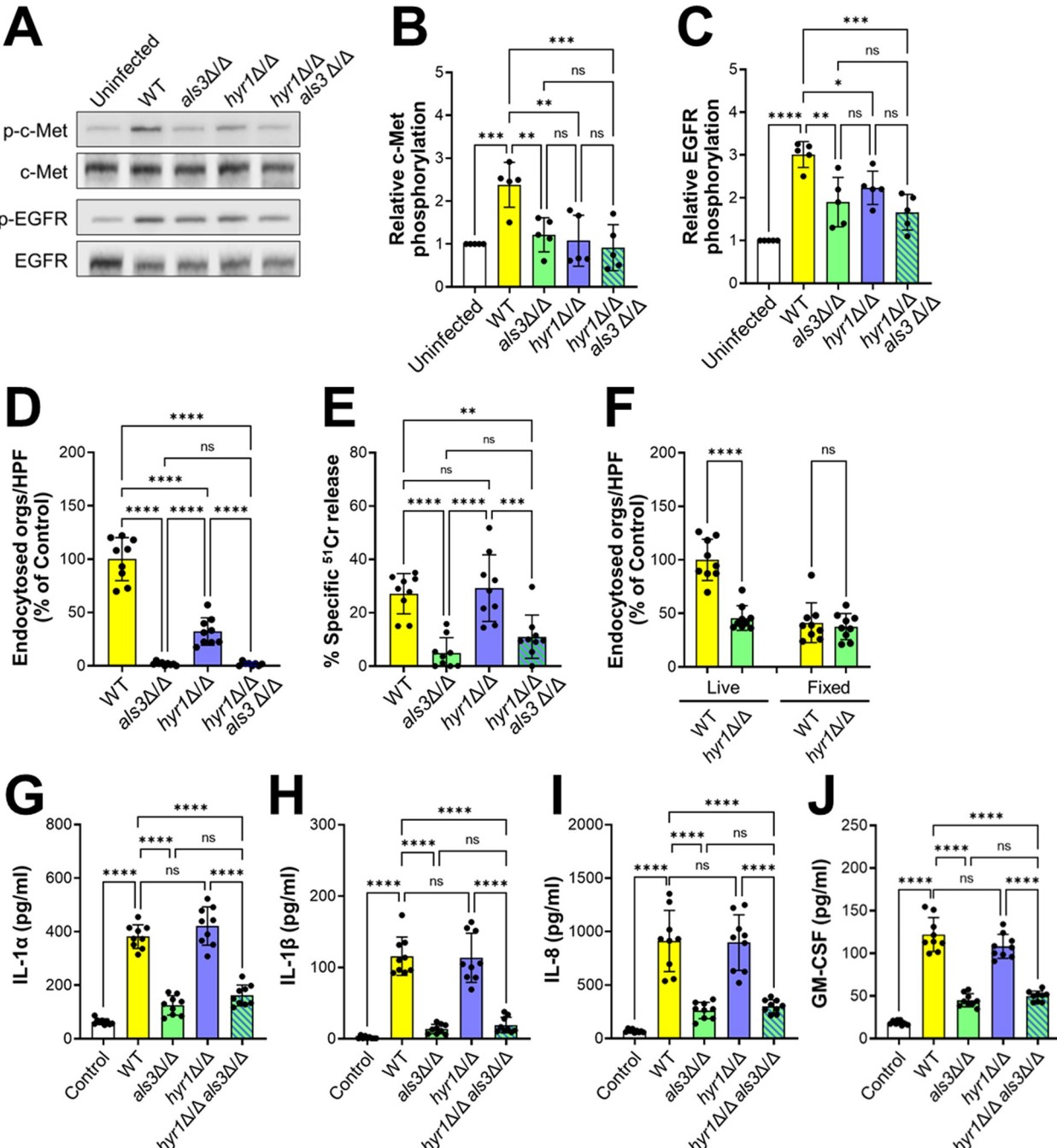

**Fig 7. Interactions of c-Met with Als3.** (A-C) Both Hyr1 and Als3 are required for phosphorylation of c-Met and EGFR. Representative immunoblots from oral epithelial cells infected with the indicated strains of *C. albicans* (A). Densitometric analysis of 5 immunoblots showing the phosphorylation of c-Met (B) and EGFR (C). (D) Epithelial cell endocytosis of the indicated strains of *C. albicans*. (E) Damage to oral epithelial cells caused by the indicated strains of *C. albicans*. (F) Invasion of live and fixed epithelial cells by the wild-type (WT) *hyr1Δ/Δ* mutant strains. (G-J) Induction of epithelial cell secretion of IL-1α (G), IL-1β (H), IL-8 (I) and GM-CSF (J) by the indicated strains of *C. albicans*. Results are mean ± SD. WT, wild type; ns, not significant; *$p < 0.05$, **$p < 0.01$, ***$p$, 0.001, ****$p < 0.0001$ (one-way ANOVA with Sidak's multiple comparisons test).

A possible explanation for these results is that although the *hyr1Δ/Δ* mutant is defective in invading epithelial cells by induced endocytosis, it may still invade these cells via active penetration. To test this possibility, we assessed the invasion of the *hyr1Δ/Δ* mutant into fixed

epithelial cells and found that it was similar to the wild-type strain (Figs 7F and S4F). Thus, Hyr1 is required for induced endocytosis, but not active penetration. This result is in contrast to what has been found for Als3, which is required for both induced endocytosis and active penetration [29], and provides a likely explanation for why Hyr1 is dispensable for induction of epithelial cell damage.

## *C. albicans* stimulates an inflammatory response in oral epithelial cells independently of Hyr1

Because invasion mediated by Als3 is required for *C. albicans* to stimulate a pro-inflammatory response by oral epithelial cells [22], we tested the capacity of the *hyr1Δ/Δ*, *als3Δ/Δ*, and *hyr1Δ/Δ als3Δ/Δ* mutants to induce the production of pro-inflammatory cytokines. As expected, epithelial cells infected with the *als3Δ/Δ* mutant released significantly less IL-1α, IL-1β, IL-8, and GM-CSF than cells infected with the wild-type strain (Fig 7G–7J). However, infection with the *hyr1Δ/Δ* mutant did not reduce cytokine production relative to infection with the wild-type strain. Also, infection with the *hyr1Δ/Δ als3Δ/Δ* double mutant did not decrease cytokine production any more than infection with the *als3Δ/Δ* single mutant. Collectively, these data indicate that Hyr1 is dispensable for *C. albicans* to induce a proinflammatory response by oral epithelial cells in vitro.

## Hyr1 reduces susceptibility to neutrophil killing independently of c-Met

Previously, it was discovered that Hyr1 enables *C. albicans* to resist killing by neutrophils [24]. We verified this result with human neutrophils using *hyr1Δ/Δ* mutants constructed in both the SC5314 and SN250 strain backgrounds (Figs 8A and S5A). Deletion of *HYR1* also rendered *C. albicans* more susceptible to killing by mouse neutrophils (Fig 8B). Similar to the *hyr1Δ/Δ* mutant, the *als3Δ/Δ* mutant had increased susceptibility to neutrophil killing and the *hyr1Δ/Δ als3Δ/Δ* double mutant was not more susceptible to killing than either single mutant (Fig 8A). Thus, both Hyr1 and Als3 reduce the susceptibility of *C. albicans* to neutrophil killing, likely by functioning in the same pathway.

To determine if Hyr1 influences susceptibility to neutrophil killing by interacting with c-Met, we tested neutrophils from Mrp8;*Met*^fl/fl^ mice, which have a neutrophil-specific deletion in c-Met [30]. Although the c-Met deficient neutrophils had reduced capacity to kill *C. albicans*, the *hyr1Δ/Δ* mutant was still more susceptible to killing by these neutrophils than the wild-type strain (Fig 8B). These results suggest that although c-Met is required for maximal neutrophil fungicidal activity, Hyr1 reduces susceptibility to neutrophil killing by a mechanism that is independent of this receptor.

## Hyr1 and Als3 are required for maximal virulence during OPC

To verify our in vitro data, we investigated the role of c-Met and Hyr1 in the pathogenesis of OPC in mice. We orally infected immunocompetent mice with the wild-type strain and the various mutants and then analyzed the phosphorylation of c-Met and EGFR in thin sections of the tongue by indirect immunofluorescence using phosphospecific antibodies. We found that there was low c-Met phosphorylation in the oral epithelium of uninfected mice and that this phosphorylation increased after infection with wild-type *C. albicans* (Fig 9A). The phosphorylation of c-Met was reduced in mice infected with either the *als3Δ/Δ* or *hyr1Δ/Δ* single mutants and was almost undetectable in mice infected with the *hyr1Δ/Δ als3Δ/Δ* double mutant. Similar results were obtained when the tongue sections were stained with a phosphospecific anti-EGFR antibody (Fig 9B). Thus, Als3 and Hyr1 are required to activate both c-Met and EGFR during OPC in mice.

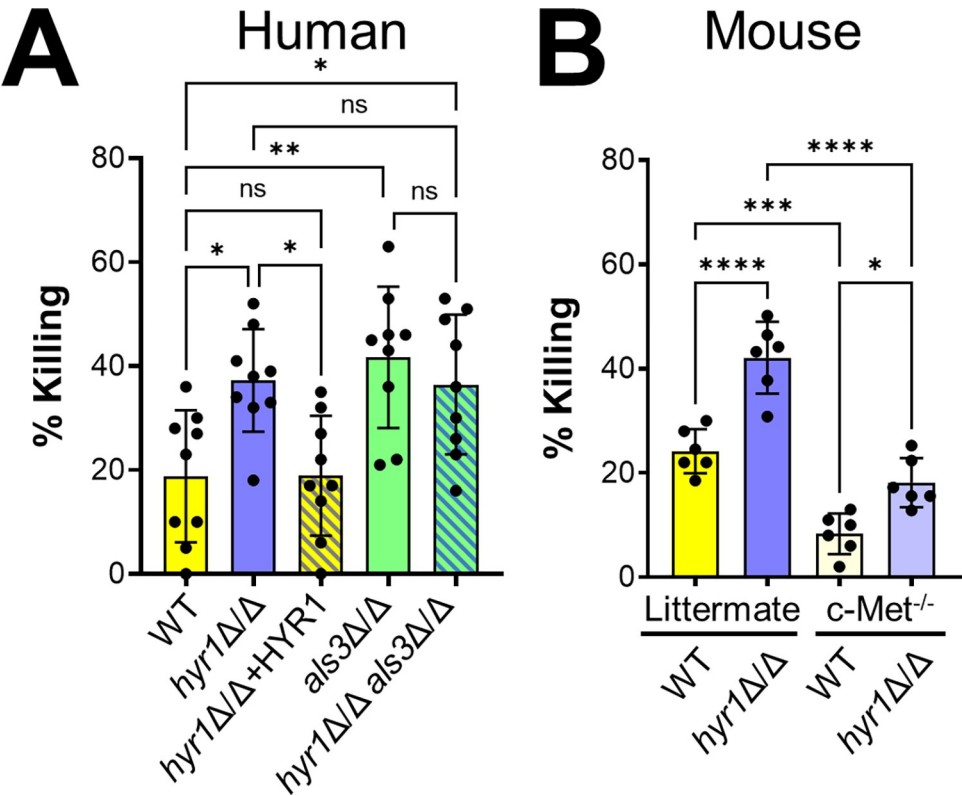

**Fig 8. Hyr1 and Als3 are required for *C. albicans* to resist killing by neutrophils.** (A) Human neutrophils were infected with the indicated *C. albicans* strains constructed in the SC5314 strain background. Results are mean ± SD of neutrophils from 3 donors, tested in triplicate. (B) The *hyr1Δ/Δ* mutant is resistant to killing by mouse neutrophils, even in the absence of c-Met. Killing of wild-type and *hyr1Δ/Δ* mutant strains in the SN250 strain background by neutrophils from Mrp8;*Met*$^{fl/fl}$ mice, which have a neutrophil-specific deletion in c-Met, and their littermates. Results are mean ± SD of neutrophils from 2 experiments performed in triplicate. ns, not significant; *$p < 0.05$, **$p < 0.01$, ***$p < 0.001$, ****$p < 0.0001$ (one-way ANOVA with Sidak's multiple comparisons test).

Next, we tested the virulence of the various mutants in immunocompetent mice using oral fungal burden as the endpoint. After 1 d of infection, the oral fungal burden of mice infected with either the *als3Δ/Δ* or *hyr1Δ/Δ* mutants was similar to that of mice infected with the wild-type strain, whereas mice infected with the *hyr1Δ/Δ als3Δ/Δ* double mutant had a significantly lower fungal burden (Fig 10A). After 2 d of infection, the oral fungal burden of mice infected with all three mutants was lower than the mice infected with the wild-type strain (Fig 10B). Collectively, these results indicated that both Hyr1 and Als3 mediate virulence during OPC.

To determine whether the virulence defect of the *hyr1Δ/Δ als3Δ/Δ* double mutant was due to reduced epithelial cell invasion or increased susceptibility to neutrophil killing, we tested the virulence of this mutant in mice in which neutrophils and monocytes were depleted with an anti-GR-1 antibody. At day 1 post-infection, the oral fungal burden of the phagocyte-deficient mice infected with the *hyr1Δ/Δ als3Δ/Δ* double mutant was similar to mice infected with the wild-type strain (Fig 10C). The absence of a virulence defect of the *hyr1Δ/Δ als3Δ/Δ* double mutant in the phagocyte-deficient mice suggests that in phagocyte-replete mice, the reduced virulence of this mutant at this time point is due mainly to its increased susceptibility to phagocyte killing. At day 2 post-infection, the oral fungal burden of the phagocyte-depleted mice infected with the *hyr1Δ/Δ als3Δ/Δ* double mutant was significantly lower than that of mice infected with the wild-type strain (Fig 10D). Thus, the attenuated virulence of the *hyr1Δ/Δ*

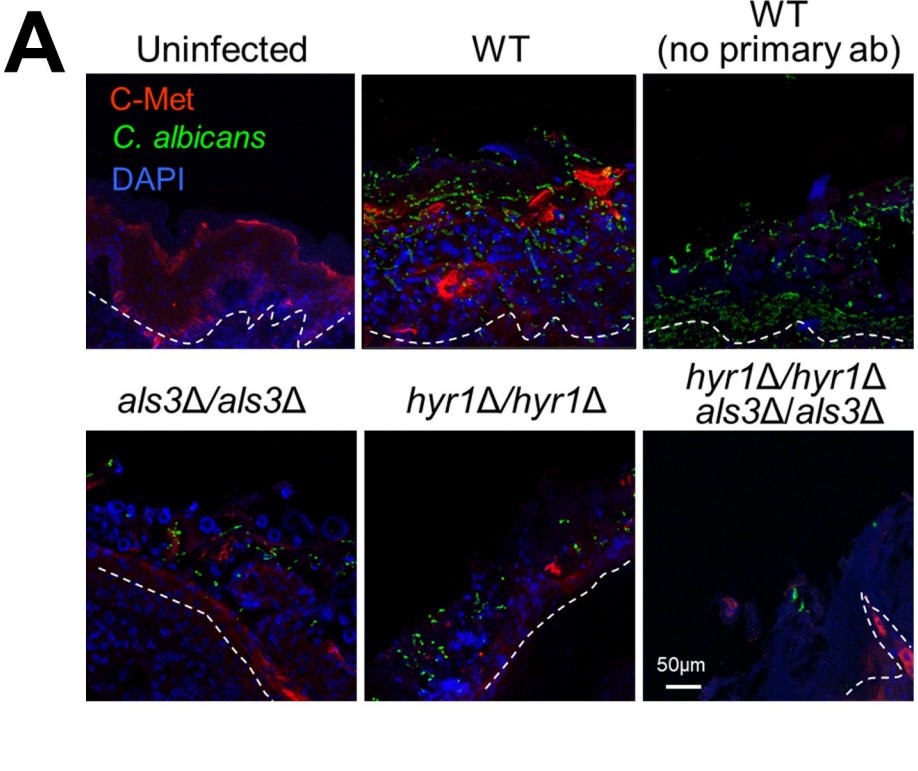

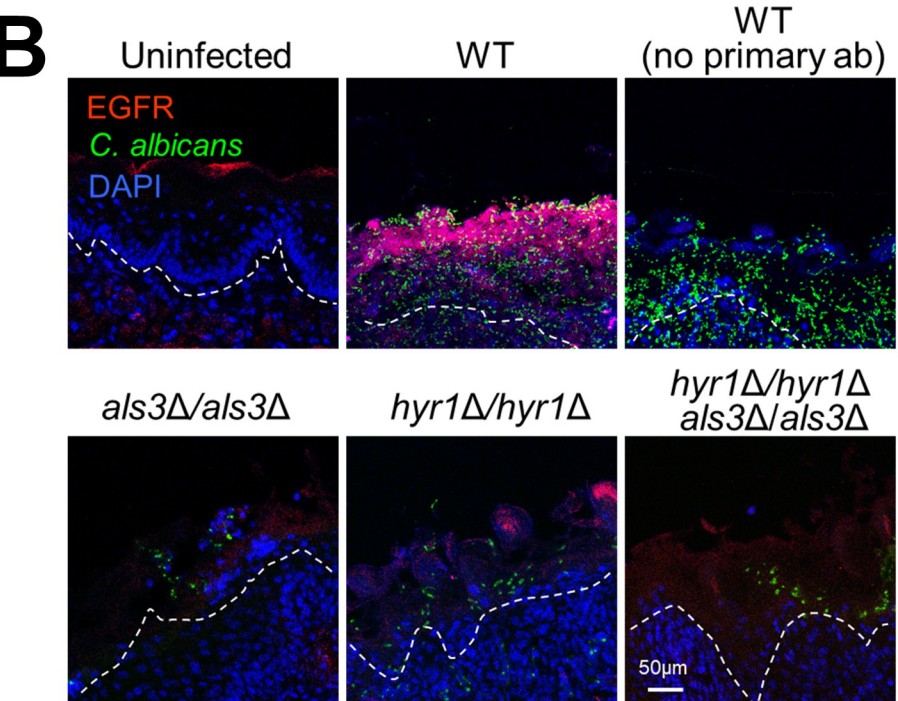

**Fig 9. Als3 and Hyr1 are required for *C. albicans* to activate c-Met and EGFR during OPC in mice.** Confocal micrographs of the tongues of immunocompetent mice after 1 d of infection with the indicated strains of *C. albicans*. The samples were stained for c-Met, *C. albicans*, and DAPI (A) or EGFR, *C. albicans*, and DAPI (B). Epithelial cells are in the region above the dotted lines. Scale bar 50 μm.

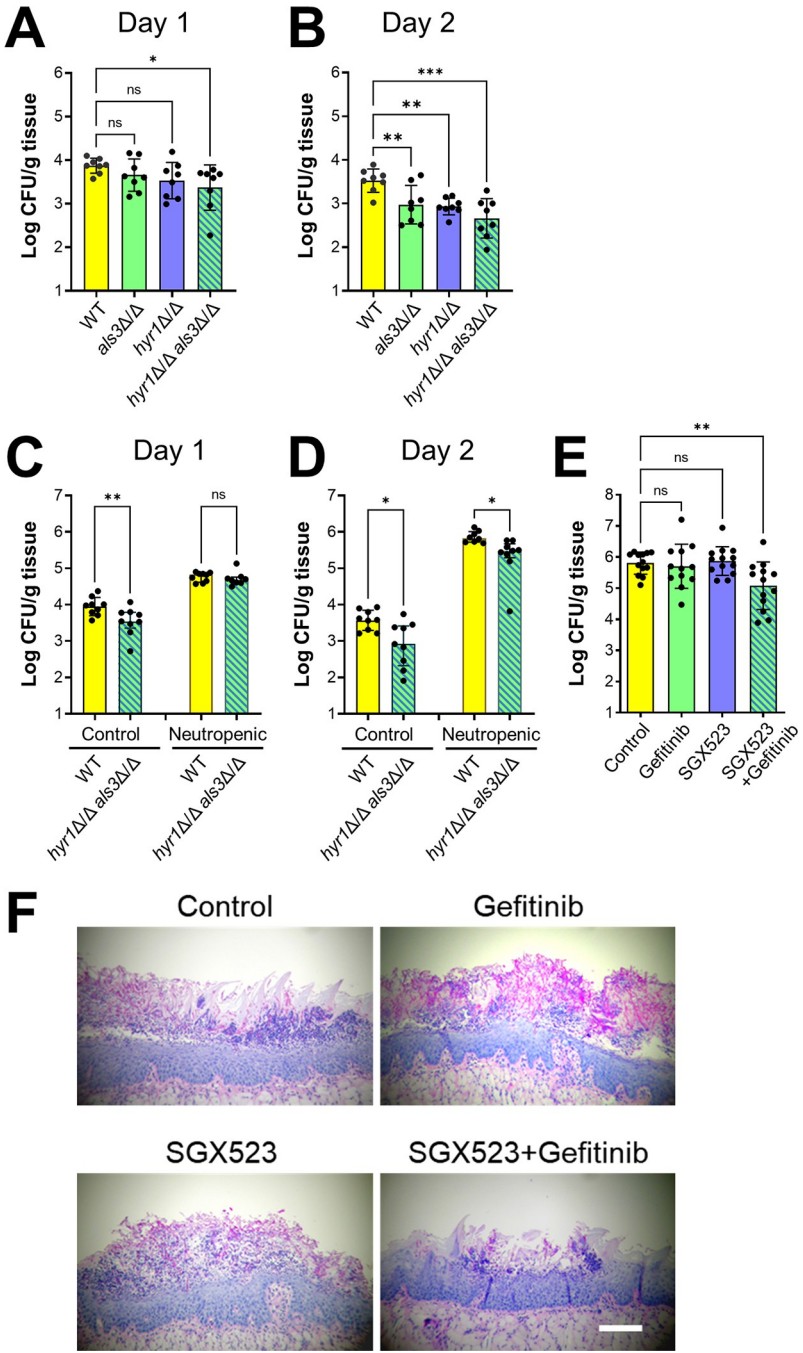

**Fig 10. Hyr1 and Als3 mediate virulence during OPC.** (A and B) Oral fungal burden of immunocompetent mice infected with the indicated *C. albicans* strains for 1 (A) and 2 (B) days. (C and D) Effects of phagocyte depletion on susceptibility to OPC, as determined by oral fungal burden after 1 (C) and 2 (D) days of infection. (E) Oral fungal burden after 5 d of infection with the wild-type strain of mice that had been immunosuppressed with cortisone acetate and treated with gefitinib and/or SGX523. (F) Histopathology of the tongues. Scale bar 200 μm. Data are the mean ± SD combined results from 2 independent experiments, each using 4 mice per *C. albicans* strain (A-D) or 8 mice per condition in one experiment and 5 mice per condition in the other (E). ns, not significant; *$p < 0.05$, **$p < 0.01$, ***$p < 0.001$, ****$p < 0.0001$ (one-way ANOVA with Sidak's multiple comparisons test).

*als3*Δ/Δ double mutant at this later time point is likely due to a phagocyte-independent mechanisms, probably reduced epithelial cell invasion.

To examine the therapeutic potential of inhibiting EGFR and c-Met, we tested the effects of gefitinib and/or SGX523 in mice that had been treated with cortisone acetate to mimic the immunosuppression that is typically seen in patients with OPC. We found that the oral fungal burden of mice treated with either gefitinib or SGX523 was similar to that of the untreated control mice, whereas the oral fungal burden of mice treated with both gefitinib and SGX523 was significantly lower (Fig 10E). Consistent with SGX523 results, otherwise immunocompetent Mrp8;*Met*^fl/fl mice with a neutrophil-specific deletion in c-Met had a similar oral fungal burden as their wild-type littermates after 2 days of infection with the wild-type strain (S5B Fig). The results with SGX523 and gefitinib were confirmed by histopathology (Fig 10F). Large fungal lesions were observed in the tongues of the control mice and those that received gefitinib or SGX523 alone, but only small lesions containing few fungal cells were visible in the mice that received combination therapy. To verify that the inhibitors blocked the phosphorylation of c-Met and EGFR, we used phosphospecific antibodies to label the tongues of immunosuppressed mice after 5 days of infection with wild-type *C. albicans*. Although the epithelium in the fungal lesions was largely destroyed at this time point, there was detectable phosphorylation of c-Met and EGFR in the infected control mice (S6 Fig). The phosphorylation of both c-Met and EGFR was substantially reduced when the mice treated with SGX523 and/or gefitinib. Thus, dual inhibition of EGFR and c-Met ameliorates OPC.

## Discussion

The data presented here indicate that *C. albicans* interacts with c-Met on oral epithelial cells and that this interaction is required for the normal endocytosis of the fungus. This conclusion is supported by our findings that infection with *C. albicans* hyphae induced c-Met autophosphorylation and that inhibition of c-Met by either siRNA or the c-Met specific kinase inhibitor SGX523 significantly reduced the endocytosis of *C. albicans*. c-Met is also known to function as a receptor for *L. monocytogenes* internalin B (InlB) and mediates the endocytosis of this organism [31]. *Helicobacter pylori* activates c-Met by injecting cytotoxin-associated gene A (CagA) into gastric epithelial cells via the type IV secretion system. CagA binds to and activates c-Met, stimulating epithelial cell motility and invasiveness, processes that are precursors to carcinogenesis [32]. Thus, three different epithelial cell pathogens have evolved independent strategies that target c-Met.

A key finding was that the combination of SGX523 and gefitinib inhibited the endocytosis of *C. albicans* more than either inhibitor alone, indicating that c-Met and EGFR/HER2 have additive effects on the endocytosis of *C. albicans*. An additive or synergistic interaction between c-Met and EGFR has been found in multiple types of cancer cells, including breast cancer, hepatomas, and glioblastomas [33–35]. In most of these cancer cells, the activation of c-Met requires EGFR and vice versa. By contrast, we found that inhibition of c-Met with SGX523 did not block *C. albicans*-induced phosphorylation of EGFR, and that inhibition of EGFR with gefitinib did not reduce phosphorylation of c-Met. These results suggest that in response to *C. albicans* infection, each of these receptors is activated independently of the other.

The current data indicate that E-cadherin is a central component of both the c-Met and EGFR endocytosis pathways. The proximity ligation and co-immunoprecipitation data showed that *C. albicans* infection cause c-Met and EGFR to form a complex with E-cadherin. Importantly, siRNA knockdown of E-cadherin inhibited *C. albicans*-induced autophosphorylation of both receptors and inhibited the endocytosis of *C. albicans* similarly to inhibition of c-Met and EGFR. While both c-Met and EGFR are known to form complexes with E-cadherin

in cancer cells, the formation of such complexes is usually associated with inhibition of c-Met or EGFR phosphorylation [36–38]. For example, in gastric cancer cells, *H. pylori* CagA activates c-Met and causes it to form a complex with E-cadherin, which inhibits c-Met phosphorylation [39]. By contrast, in the non-tumorgenic HaKat keratinocyte cell line, assembly of E-cadherin-containing adherens junctions activates EGFR in a ligand-independent manner [40]. We speculate that the interactions of *C. albicans* with E-cadherin may activate both c-Met and EGFR by a similar process.

Previously, we identified two *C. albicans* invasins, Als3 and Ssa1 that mediate invasion of oral epithelial cells by interacting with E-cadherin and EGFR [16,17,19]. In the current work, we discovered that Hyr1 also functions as an invasin that is required for *C. albicans* to activate both EGFR and c-Met in vitro and in the mouse model of OPC. Although Hyr1 and Als3 interacted with the same epithelial cell receptors, deletion of *HYR1* and *ALS3* had different effects on the interactions of *C. albicans* with epithelial cells. The *hyr1Δ/Δ* mutant stimulated less epithelial cell endocytosis but induced wild-type levels of damage and cytokine release. By contrast, the *als3Δ/Δ* mutant induced very little endocytosis, damage, or cytokine release. One potential explanation for these results is that although the *hyr1Δ/Δ* mutant has reduced receptor mediated endocytosis, it can still invade epithelial cells by active penetration, whereas the *als3Δ/Δ* mutant is defective in both modes of epithelial cell invasion. In contrast to the *als3Δ/Δ* mutant, it is likely that the *hyr1Δ/Δ* mutant can still form an invasion pocket via active penetration into which candidalysin can accumulate to sufficient levels to stimulate and damage oral epithelial cells.

Hyr1 has been found to contribute to virulence by enabling *C. albicans* to resist being killed by neutrophils [24]. Our data support this finding and also suggest that Als3 induces resistance to neutrophil killing by functioning in the same pathway as Hyr1. Previously, we found that *C. glabrata* cells that were engineered to express *C. albicans* Als3 had enhanced resistance to neutrophil killing, thus supporting this novel function of Als3 [41]. How Hry1 and Als3 mediate resistance to killing is currently unknown. Although our data with c-Met deficient neutrophils indicate that c-Met is required for maximal killing of *C. albicans*, they also show that Hyr1 mediates resistance to neutrophil killing independently of c-Met. We speculate that Hyr1 and Als3 may interact with receptors other than c-Met on the surface of neutrophils to down-regulate their fungicidal activities.

The results of the current studies indicate that Hyr1 is required for maximal virulence in during OPC in immunocompetent mice. It has previously been found that Hyr1 also required for maximal virulence in an immunosuppressed mouse model of OPC [25]. As predicted by our in vitro results, we found that Hyr1 and Als3 mediate virulence during OPC in mice by functioning in the same pathway. The results with phagocyte-depleted mice also suggest that after 1 day of infection, Hyr1 and Als3 contribute to virulence by reducing susceptibility to phagocyte killing, whereas after 2 days of infection, these surface proteins contribute to virulence by a process that is independent of phagocytes, likely by enhancing epithelial cell invasion. We also found that in immunosuppressed mice, the combined inhibition of EGFR and c-Met ameliorated the severity of OPC, suggesting that strategies to block these host cell receptors hold promise to prevent or treat this infection.

In summary, this work identifies c-Met as a novel oral epithelial cell receptor that forms a complex with EGFR and E-cadherin to mediate the endocytosis of *C. albicans*. In addition to interacting with Als3, this complex interacts with Hyr1 and induces epithelial cell endocytosis, thus demonstrating that Hyr1 functions as an invasin. Previous studies by our group indicate that Hyr1 is a promising vaccine target for prevention of both *C. albicans* and *Acinetobacter baumanii* infections [42–44]. The current data suggest that the host cell targets of Hyr1, EGFR and c-Met are also potential therapeutic targets.

## Materials and methods

### Ethics statement

All animal work was approved by the Institutional Animal Care and Use Committee at the Lundquist Institute for Biomedical Innovation at Harbor-UCLA Medical Center. After written informed consent was obtained, blood was drawn from normal volunteers via a protocol approved by the Institutional Review Board of the Lundquist Institute for Biomedical Innovation at Harbor-UCLA Medical Center.

### Host cells

The OKF6/TERT-2 oral epithelial cell line was provided by J. Rheinwald (Dana-Farber/ Harvard Cancer Center, Boston, MA) [21], and cultured as described previously [17]. The NIH/3T3 cell line expressing human EGFR and HER2 [45] was provided by Nadege Gaborit (Institut de Researche en Cancérologie de Montpellier, France) and grown as described [17].

### Fungal cells

The various *C. albicans* strains (S2 Table) were grown in YPD broth at 30˚C in a shaking incubator for 18 h, after which the cells were pelleted by centrifugation and washed twice in PBS. Yeast cells were suspended in PBS and counted with a hemacytometer. Germ tubes were produced by adding the yeast to RPMI 1640 broth at a final concentration of $2x10^6$ organisms per ml and incubating them for 90 min at 37˚C. The resulting germ tubes were removed from the petri dish with a cell scraper, resuspended in PBS, sonicated briefly, and then enumerated with a hemacytometer.

### Mice

C57Bl/6 mice were purchased from Jackson Laboratories. Mrp8;*Met*^fl/fl mice were a generous gift from Massimiliano Mazzone [30] and bred at the Lundquist Institute. Before the experiments, the wild-type and Mrp8;*Met*^fl/fl mice were co-housed for 2 weeks, and then used at age 10–12 weeks. Both male and female mice were used in the experiments.

### Strain construction

Strains MC355, MC374, and MC502 were generated from strain SC5314 utilizing the transient CRISPR-Cas9 system [46] with recyclable markers [47]. In brief, the SC5314 wild-type strain was made to be auxotrophic for histidine production via homozygous deletion of *HIS1* via integration of the recyclable nourseothricin resistance marker, *NAT1* [48]. The *his1Δ::r3NAT1r3* strain was again transformed in this study to delete the *HYR1* coding sequence using sgRNAs targeting *HYR1* and the recyclable *NAT1* marker in addition to a recyclable *Candida dubliniensis* (*C.d.*) *HIS1* marker flanked with 80-300bp homology to the 5'and 3'*HYR1* regions. The resulting strain (MC355) with genotype *hyr1Δ::r1C.d.HIS1r1, his1Δ::r3* was transformed again to ectopically complement *HYR1* by homozygous integration of a WT copy of the *HYR1* coding region at the *MDR1* locus. PCR-amplified *HYR1* with 1553bp 5'and 661bp 3'flanking sequence concatenated with a selectable *NAT1* marker were inserted at the *MDR1* locus [49,50]. The resulting complemented strain (MC502) had the genotype: *hyr1Δ::r1C.d.HIS1r1, his1Δ::r3, mdr1Δ::NAT1-HYR1*. MC355 was then used to additionally delete *ALS3* by a recyclable *NAT1* marker flanked by 80-300bp homology to the *ALS3* up and downstream regions and a sgRNA targeting the *ALS3* coding sequence. The resulting strain (MC374) had the genotype: *hyr1Δ::r1C.d.HIS1r1, his1Δ::r3, als3Δ::r3NAT1r3*. Transformations using the *NAT1* or *HIS1* markers were plated on either YPD + nourseothricin or complete

synthetic medium (CSM) lacking histidine respectively. All strain genotypes were verified by PCR amplification of the desired marker from the target locus. The PCR primers are listed in S3 Table.

## Inhibitors and stimuli

Gefitinib (#S1025, Selleck Chem) was dissolved in DMSO at concentration of 100 mM, and diluted further to 1 μM in KSF medium (#17005042, Thermo Fisher Scientific) without supplements for final use. SGX-523 (#S1112, Selleck Chem) was dissolved in DMSO and diluted to 200 nM in KSF medium without supplements for final use. DMSO was used as control and diluted in the same manner. The anti-EGFR antibody (Erbitux, Bristol-Myers Squibb) was diluted to 10 μg/ml in KSF medium without supplements for use. EGF (#10450–013, Thermo Fisher Scientific) was diluted in PBS containing 0.1% BSA at a concentration of 2 μg/ml, and diluted to a final concentration of 20 or 40 ng/ml in KSF medium without supplements.

## Immunofluorescence

The immunofluorescence studies and proximity ligation assays were performed using minor modifications of our previously described methods [51,52]. OKF6/TERT cells were seeded onto fibronectin coated circular glass coverslips in 24-well tissue culture plates and incubated at 37˚C in 5% $CO_2$ overnight. The cells were infected with 3 x $10^5$ _C. albicans_ germ tubes in KSF medium without supplements for 20 min. After aspirating the medium, the cells were fixed with 4% paraformaldehyde, rinsed with PBS, and permeabilized with 0.1% Triton X-100 (Sigma-Aldrich) in PBS. The cells were blocked with 5% goat serum in PBS, and then stained with mouse anti-c-Met (#370100, Invitrogen), rabbit anti-EGFR (#GTX121919, Genetex), or anti-E-cadherin antibodies (#GTX100443, Genetex). Control slides were stained with rabbit igG antibodies (#02–6102, Life Technologies) and mouse igG antibodies (#MAB002, R&D systems) followed by the appropriate fluorescent labeled secondary antibodies. _Candida albicans_ were stained with anti _candida_ Alexa Fluor 488. The coverslips were mounted inverted using antifade mounting medium and imaged by confocal microscopy. Multiple images were obtained along the z-axis and stacked using LAS X software (Leica Microsystems).

For the proximity ligation assay, the cells were processed similarly except that they were infected with germ tubes of a GFP-expressing strain of _C. albicans_ for 20 min. The following pairs of primary antibodies were used: mouse anti-c-Met (#370100, Invitrogen) and rabbit anti-E-cadherin antibodies (#GTX100443, Genetex), mouse anti-EGFR (#SC-101, Santa Cruz Biotechnology) and rabbit anti-E-cadherin (#GTX100443, Santa Cruz Biotechnology), mouse anti-c-Met and rabbit anti-EGFR (#GTX121919, Genetex). The control slides were incubated with anti rabbit igG (#02–6102, Life Technologies), and anti mouse igG (#MAB002, R&D systems). The interactions between the two labeled proteins were detected using the Duolink in Situ Red Starter Kit Mouse/Rabbit (#DUO92101-1kit, Sigma-Aldrich) according to the manufacturer's instructions.

## Protein phosphorylation

The capacity of the various _C. albicans_ strains to induce phosphorylation of EGFR and c-Met in the presence and absence of inhibitors was determined as described previously [18]. Briefly OKF6/TERT cells were seeded onto 24 well plates and incubated overnight in KSF medium without supplements. The next morning, the medium was aspirated and replaced with either fresh medium alone or containing gefitinib and/or SGX-523. When inhibitors were used, control cells were incubated with KSF medium containing a similar volume of the DMSO diluent. After 1 h, the cells were infected with 1x$10^6$ _C. albicans_ germ tubes in the presence of the

inhibitors and incubated for 20 min. Next, the medium was aspirated and the epithelial cells were lysed with 2X SDS loading buffer (#BP-111R, Boston Bioproducts, Inc.) containing phosphatase/protease inhibitors (# A32959, Thermo Fisher Scientific), and PMSF (#P7626, Sigma-Aldrich). After denaturing the samples at 90˚C for 2 minutes, the lysates were clarified by centrifugation. The proteins were separated by SDS-PAGE and transferred to PVDF membranes. The phosphorylated proteins were detected by probing the membranes with an anti-phospho-c-Met antibody (Tyr1234/1235, #3077, Cell Signaling Technology) or an anti-phospho-EGFR antibody (Tyr1068, #2234, Cell Signaling Technology). Next the blots were stripped and total c-Met was detected with an anti-met antibody (# 8198, Cell Signaling Technology) and total EGFR was detected with an anti-EGFR antibody (#4267, Cell Signaling Technology). The blots were developed using enhanced chemiluminescence, imaged with a digital imager, and quantified using Image Studio Lite software. Each experiment was repeated at least four times.

### siRNA

As previously described [51], OKF6/TERT2 cells were grown in 6 well plates to 80% confluency and transfected with 40 pmole of c-Met siRNA (#SC-29397, Santa Cruz Biotechnology) or E-cadherin siRNA (#SC-35242, Santa Cruz Biotechnology) using Lipofectamine 2000 (#11668027, ThermoFisher Scientific) following the manufacturer's instructions. After 24 h, the cells were trypsinized, seeded onto fibronectin coated glass coverslips, and incubated for another 24 h before use. The extent of protein knockdown was determined by immunoblotting with an anti-c-Met antibody (#8198, Cell Signaling Technology), anti-E-cadherin antibody (#3195, Cell Signaling Technology), anti-EGFR antibody (#4267, Cell Signaling technology), and anti-β-actin antibody (#A5441, Sigma-Aldrich).

### Adherence and endocytosis assay

The number of fungal cells that were endocytosed by and cell-associated with the host cells was determined using our differential fluorescence assay as described [16]. For OKF6/TERT2 cells, the inoculum was $1x10^5$ *C. albicans* germ tubes and the incubation period was 1 h. For NIH/3T3 cells, the inoculum was $1x10^5$ *C. albicans* yeast and the incubation period was 1.5 h. To measure active penetration, the OKF6/TERT2 cells were fixed with 4% paraformaldehyde for 15 min, rinsed extensively with HBSS and then infected with $1 X 10^5$ *C. albicans* germ tubes for 2.5 h. In all experiments, at least 100 organisms were scored per coverslip. The experiments were repeated three times in triplicate.

### Lentivirus production and transduction

NIH/3T3 cells were engineered to express human c-Met by lentivirus transduction. To construct the pLenti-EF1A-EGFP-Blast and pLenti-EF1A-hcMet-BLAST transfer vectors, GFP or human c-MET cDNA were PCR amplified from plasmid pLenti-MetGFP (Addgene #37560) and seamlessly cloned into the BamHI/XbaI sites of vector pLenti-spCas9-Blast (Addgene #52962). The lentiviruses were packaged by transfecting HEK293T cells with plasmids psPAX2 (#12260, Addgene), pCMV-VSVG (#8454, Addgene), and each transfer vector using the X-tremeGENE 9 DNA transfection reagent (#6365787001, Sigma-Aldrich) according to the manufacturer's instructions. The viruses were collected 60 h after transfection. NIH/3T3 cells were transduced with each lentivirus in the presence of polybrene (#SC134220, Santa Cruz Biotechnology) and the transduced cells were selected by adding blasticidin (#A1113903, Gibco-BRL) to the medium 2 d later. To verify that the transduced cells expressed human c-Met, immunoblots of cell lysates were probed with antibodies against c-Met (#8198, Cell Signaling Technology) and GAPDH (#5174, Cell Signaling Technology).

## Immunoprecipitation

The immunoprecipitation experiments were performed using a minor modification our previously described method [51]. Briefly, oral epithelial cells were infected with *C. albicans* germ tubes for 20 min and then lysed with n-octyl-β-glupyranoside (#97061–760, Sigma-Aldrich). After preclearing the cell lysates with protein A/G beads (#SC 2003, Santa Cruz Biotechnolgy), the lysates were incubated with an anti-EGFR (SC-101, Santa Cruz Biotechnology) or anti-c-Met (#370100, Invitrogen) antibodies for 1 h at 4˚C, and precipitated with protein A/G beads for 2 h at 4˚C. The beads were collected by centrifugation and washed 3 times with 1.5% octyl-β-glucopyranoside in the present of proteinase inhibitors. The proteins were eluted with 2X SDS PAGE loading buffer, denatured at 90˚C for 2 min, and then separated by SDS PAGE. Specific epithelial cell proteins were detected by immunoblotting with anti-c-Met (#370100, Invitrogen), anti-EGFR (#2234, Cell Signaling technologies), and anti-E-cadherin (#3195, Cell Signaling Technologies). All experiments were repeated at least 4 times.

## Far western blotting

Protoplasts of yeast-phase and germinated *C. albicans* strains were prepared by suspending the organisms in pretreatment buffer containing 10mM tris-HCl, pH9.0, 5mM EDTA, and 1% β-mercaptoethanol to a density of $1–2 \times 10^8$ cells/ml and incubating them at 28˚C for 30 min with gentle shaking. The cells were collected by centrifugation, washed with 1M sorbitol and then resuspended in 1M sorbitol containing 30 µl glusulase per ml (# EE154001EA, Perkin Elmer) to a density of $5 \times 10^8$ cells/ml. The cells were incubated at 28˚C with gentle shaking for approximately 1 h until 90% of cells had converted to protoplasts. After harvesting the protoplasts by centrifugation, they were washed 3 times with 1M sorbitol and resuspended in RPMI 1640 broth containing protease inhibitor cocktail (Sigma-Aldrich) to final concentration of $3 \times 10^8$ cells/ml. The protoplasts were allowed to regenerate by incubation at 37˚C with gentle shaking for 90 minutes, after which the cells were pelleted by centrifugation and the supernatants were collected. The concentration of the proteins in the supernatants was measured using the Bradford assay. For far Western blotting, 15 µg of protein separated by non-denaturing SDS-PAGE and transferred to a PVDF membrane. After blocking with 5% milk/TBST for 1 hr, the membrane was incubated with recombinant c-Met (# MET-H5227, ACROBiosystems) followed by an anti human c-Met antibody (# AF276, R&D systems). The bands were visualized by enhanced chemiluminescence. In parallel 10µg of protein was separated by non-denaturing SDS-PAGE and the gel was stained with coomassie blue. The bands of interest were excised and the proteins in them were sequenced by mass spectrometry.

## Epithelial cell damage

The extent of damage to the oral epithelial cells caused by different *C. albicans* strains was measured by a $^{51}$Cr release assay in 96-well plates as described previously [16,22]. The inoculum was $2.5 \times 10^5$ cells per well and the incubated period was 8 h. Each experimenta was repeated three times in triplicate.

## Cytokine assay

Oral epithelial cells in 24 well tissue culture plates were incubated overnight in KSF medium without supplements. The next morning, the cells were infected with $1.5 \times 10^6$ *C. albicans* yeast in the same medium. After 8 h, the medium above the cells was collected, centrifuged to remove cell debris, and stored at -80˚C. The concentration of IL-1α, IL-1β, IL-8, and GM-CSF in the samples was measured using the Luminex Multiplex panel (# LXSAHM-04, R&D

systems) according to the manufacturer's instructions. The experiments were repeated three times in triplicate.

## Neutrophil killing assay

To test the susceptibility of *C. albicans* strains to killing by human neutrophils, blood was collected from healthy volunteers by venipuncture and mixed with $K_3$EDTA (#E-0270, Sigma-Aldrich). The donors were two females and one male. The neutrophils were isolated using Lympholyte-Poly Cell Separation Media (#CL5070, Cedarlane) following the manufacturer's instructions. They were washed once with HBSS without $Ca^+/Mg^+$ (#21-022-CV, Corning), suspended in RPMI 1640 medium with L-glutamine (#9161, Irvine Scientific) containing 10% pooled human serum (#100–110, Gemini Bioproducts, Inc.) and enumerated using a hemacytometer. For the killing assay, the neutrophils were incubated with *C. albicans* yeast at the ratio of 1:1 in polypropylene tubes at 37°C. As a control, an equal number *C. albicans* cells was incubated without neutrophils in parallel. After 3 h, the neutrophils were lysed by adding sterile water to the tube, followed by sonication. The number of viable organisms was determined by quantitative culture.

The susceptibility of the *C. albicans* strains to killing by mouse neutrophils was determined similarly. Neutrophils were purified from bone marrow cells using negative magnetic bead selection (MojoSort, BioLegend) as we have done before [53]. In brief, bone marrow cells from a male and a female mouse were flushed from femurs and tibias using sterile RPMI 1640 medium supplemented with 10% FBS and 2 mM EDTA. After washing the cells with 1X Mojo-Sort buffer (1X PBS, 0.5% BSA, 2mM EDTA), the neutrophils were isolated according to the manufacturer's instructions. These neutrophils had > 95% purity and > 90% viability as determined by flow cytometry. For the killing assay, the mouse neutrophils were incubated with *C. albicans* yeast at the ratio of 1:20 in RPMI 1640 medium containing 2% heat-inactivated mouse serum (#S3509, Sigma-Aldrich) at 37°C for 3 h.

## Virulence studies

The virulence of the various *C. albicans* strains and the effects of gefitinib and SGX523 were determined using our standard mouse model of oropharyngeal candidiasis [17,22,54]. All studies were performed using male Balb/c mice that were randomly assigned to the different experimental groups. For studies with immunocompetent or phagocyte-depleted mice, the animals were inoculated with calcium alginate swabs that had been soaked in HBSS containing $2x10^7$ organisms/ml and for experiments with mice that had been immunosuppressed with cortisone acetate, the animals were inoculated with calcium alginate swabs that had been soaked in HBSS containing $1x10^6$ organisms/ml. These mice were administered gefitinib and/or SGX523 by adding it to powdered mouse chow at final concentrations of 200 ppm and 120 ppm, respectively, starting at day -1 relative to infection. To deplete the mice of phagocytes, they were administered 80 µg of an anti-GR-1 antibody (#BE0075; clone RB6-8C5, Bio X Cell) intraperitoneally on day -1 relative to infection. Control mice were injected with a similar dose of an isotype control antibody (#BE0090, Clone LTF-2, Bio X Cell). The mice were sacrificed after 1, 2 or 5 days of infection, depending on the experiment, after which the tongues were excised, weighed, and quantitatively cultured.

## Immunohistochemistry

To assess the phosphorylation of c-Met and EGFR in vivo, immunocompetent mice were infected with the various *C. albicans* strains as described above. After 1-d of infection, the mice were sacrificed, and the tongues were excised, snap frozen, and embedded into OTC. Thin

sections were prepared and transferred to glass slides. The samples were air dried, fixed in 100% methanol, rinsed, and then blocked with 5% goat serum in PBS. The slides were incubated with a rabbit anti-phospho-c-MET antibody (Tyr1003, #MBS9600900, My Biosource Inc.) or control rabbit IgG (#026102, Invitrogen) followed an Alexa Fluor 568-conjugated goat anti-rabbit antibody. The *C. albicans* cells were labeled with an anti-*Candida* antibody conjugated with Alexa Fluor 488 and the nuclei were labeled with DAPI. Phosphorylated EGFR was detected similarly, except that the slides were incubated with an anti-rabbit phosho-EGFR conjugated with phycoerythrin (Tyr1068, #14565, Cell Signaling Technologies) and control slides were stained with rabbit IgG conjugated with phycoerthrin (#5742, Cell Signaling Technologies). The slides were imaged by confocal microscopy, and z-stacks were combined using LAS X software.

## Supporting information

**S1 Fig.** (A) Immunofluorescent image of the OKF6/TERT-2 oral epithelial cell line infected with *C. albicans* and stained with control mouse IgG. Arrows indicate the organism in the magnified inset. Scale bar 10 μm. (B and C) Effects of knockdown of c-Met with siRNA (B) or inhibition of c-Met signaling with SGX523 (C) on the number of *C. albicans* cells that were associated (adherent and endocytosed) with the OKF6/TERT-2 oral epithelial cell line. (D) Immunoblot showing the effects of control and c-Met siRNA on the levels of the indicated oral epithelial cell proteins. (E) Effects of hepatocyte growth factor (HGF) treatment on the number of *C. albicans* cells that were associated with oral epithelial cells. (F) Immunoblot showing the phosphorylation of c-Met induced by a 20-min exposure to the indicated amounts of HGF. Results in (B, C, and E) are mean ± SD of 3 experiments performed in triplicate. ns, not significant (two-way Student's t test [B and C] or one-way ANOVA with Sidak's multiple comparisons test [E]).
(PDF)

**S2 Fig.** (A and B) Effects of SGX523 and gefitinib on invasion (A) and adherence (B) of *C. albicans* to live and paraformaldehyde-fixed oral epithelial cells. (C) Effects of SGX523 and/or gefitinib on the number of *C. albicans* cells that were associated with oral epithelial cells. (D and E) The number of *C. albicans* cells that were associated with wild-type NIH/3T3 cells (control) or cells expressing human c-Met (D) or cells expressing the human epidermal growth factor (EGFR) and HER2 or human c-Met, EGFR, and HER2 (E). (F and G) Effects of siRNA knockdown of E-cadherin in combination with SGX523 (F) or gefitinib (G) on the number of *C. albicans* cells that were associated with oral epithelial cells. Results are mean ± SD of 3 experiments performed in triplicate. $*p < 05$, $**p < 0.01$, $***p$, 0.001, $****p < 0.0001$, ns; not significant (one-way ANOVA with Sidak's multiple comparisons test [A-C, F, G] or two-tailed Student's t test [D and E]).
(PDF)

**S3 Fig.** (A) Proximity ligation assay performed with mouse and rabbit IgG as a negative control. Scale bar 10 μm. (B-D) Co-immunoprecipitation experiments in oral epithelial cells transfected with control or E-cadherin siRNA and then infected with *C. albicans* for 20 min. Representative immunoblots of proteins obtained by immunoprecipitation with an anti-EGFR antibody (B). Densitometric analysis of 5 immunoblots (C and D). Results are mean ± SD. $*p < 0.05$, $***p < 0.001$, $****p < 0.0001$, ns; not significant (one-way ANOVA with Sidak's multiple comparisons test).
(PDF)

**S4 Fig.** (A) Number of cells of the indicated *C. albicans* strains that were associated with oral epithelial cells. (B-D) Ece1 is dispensable for inducing the phosphorylation of c-Met but not EGFR in oral epithelial cells. Representative immunoblots (B). Densitometric analysis of 5 immunoblots showing the phosphorylation of c-Met (C) and EGFR (D) induced by the indicated strains of *C. albicans*. Results are mean ± SD. (E) Number of cells of the indicated *C. albicans* strains that were associated with oral epithelial cells. (F) Number of cells of the indicated *C. albicans* strains that were associated with live and paraformaldehyde fixed oral epithelial cells. Results in (A, E and F) are the mean ± SD of three experiments, each performed in triplicate. $^{*}p < 05$, $^{**}p < 0.01$, $^{****}p < 0.0001$, ns; not significant (one-way ANOVA with Sidak's multiple comparisons test [B-D] or two-tailed Student's t test [E and F]).
(PDF)

**S5 Fig.** (A) Human neutrophils were infected with the indicated *C. albicans* strains constructed in the SN250 strain background. Results are mean ± SD of neutrophils from 5 donors, tested in triplicate. (B) Oral fungal burden of otherwise immunocompetent Mrp8;*Met*^fl/fl mice, which have a neutrophil-specific deletion in c-Met (c-Met$^{-/-}$) and their wild-type littermates after 2 days of infection with *C. albicans* SC5314. Results are combined data from 2 experiments with a mixture of male and female mice. ns, not significant; $^{**}p < 0.01$ (two-sided Student's t test).
(PDF)

**S6 Fig. Confocal micrographs of the tongues of mice immunosuppressed with cortisone acetate and then infected for 5 days with *C. albicans* strain SC5314 (WT) in the presence of SGX523 and/or gefitinib.** The specimens were stained for c-Met, *C. albicans*, and DAPI (A) or EGFR, *C. albicans*, and DAPI (B). Epithelial cells are in the region above the dotted lines. Scale bar 50 μm.
(PDF)

**S1 Table.** *C. albicans* **proteins identified by far-Western blotting with recombinant c-Met.**
(XLSX)

**S2 Table. List of *Candida albicans* strains used in the work.**
(PDF)

**S3 Table. Oligonucleotides used in the experiments.**
(XLSX)

## Acknowledgments

We thank Adam Diab for his assistance with tissue culture.

## Author Contributions

**Conceptualization:** Aaron P. Mitchell, Scott G. Filler.

**Formal analysis:** Norma V. Solis, Scott G. Filler.

**Funding acquisition:** Marc Swidergall, Aaron P. Mitchell, Scott G. Filler.

**Investigation:** Quynh T. Phan, Norma V. Solis, Max V. Cravener, Jianfeng Lin, Manning Y. Huang, Hong Liu.

**Methodology:** Norma V. Solis, Marc Swidergall, Jianfeng Lin, Manning Y. Huang, Hong Liu, Shakti Singh, Ashraf S. Ibrahim, Aaron P. Mitchell.

**Project administration:** Scott G. Filler.

**Resources:** Shakti Singh, Ashraf S. Ibrahim, Massimiliano Mazzone.

**Supervision:** Aaron P. Mitchell, Scott G. Filler.

**Writing – original draft:** Quynh T. Phan, Norma V. Solis, Jianfeng Lin, Scott G. Filler.

**Writing – review & editing:** Quynh T. Phan, Marc Swidergall, Jianfeng Lin, Ashraf S. Ibrahim, Massimiliano Mazzone, Aaron P. Mitchell, Scott G. Filler.

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
