## [Decision Letter · Decision Letter 0]

17 May 2023

Dear Dr. Filler,

Thank you very much for submitting your manuscript "Candida albicans stimulates formation of a multi-receptor complex that mediates epithelial cell invasion during oropharyngeal infection" for consideration at PLOS Pathogens. As with all papers reviewed by the journal, your manuscript was reviewed by members of the editorial board and by several independent reviewers. The reviewers appreciated the attention to an important topic. Based on the reviews, we are likely to accept this manuscript for publication, providing that you modify the manuscript according to the review recommendations.

Sincerely,

Bruce S Klein

Academic Editor

PLOS Pathogens

Alex Andrianopoulos

Section Editor

PLOS Pathogens

Kasturi Haldar

Editor-in-Chief

PLOS Pathogens

orcid.org/0000-0001-5065-158X

Michael Malim

Editor-in-Chief

PLOS Pathogens

orcid.org/0000-0002-7699-2064

Reviewer Comments (if any, and for reference):

Reviewer's Responses to Questions

**Part I - Summary**

Reviewer #1: In this paper, Phan and colleagues are investigating the connection between two fungal cell wall proteins and endocytosis via the c-Met, EGFR, and e-Cadherin pathways. They use a combination of knockdown, silencing, and drug inhibition approaches to define the specific functions of c-MET, EGFR, and e-cadherin during C. albicans endocytosis. This host-focused data is solid, and each conclusion is supported by multiple lines of evidence. To investigate the fungal proteins involved in endocytosis, they focus on Als3 and Hyr1, using mutant strains and some complemented strains. The fungal data is less solid, as some of the phenotypes of the fungal mutants do not show a clear connection to the host receptors. Overall, this is an important contribution to the field, with some places where a few additional experiments would increase the author’s ability to make these conclusions.

Reviewer #2: This a very interesting study that provides new data about the function of Hyr1 for C. albicans receptor mediated endocytosis. The hypothesis upon upon which this manuscript is predicated is that oral epithelial cells express c-Met, E-cadherin and EFGR/HER2 receptors that become associated with each other upon association of C. albicans hyphae producing an activated complex that results in endocytosis of the fungal cell. Data supporting this model are that c-Met and EFGR are phosphorylated upon exposure to germinated C. albicans, and that inhibition of expression of these receptors by siRNA or chemical blockers reduced endocytosis. These receptors are shown to be associated following exposure to C. albicans by coIP. To identify hyphal proteins that may be involved in these interactions, Als3 and SSa1 are examined and found to be needed for activation of c-MET and EFGR. Interestingly Hyr1 is also identified as a likely partner by far Western and was then shown be needed for receptor activation and endocytosis. Analyses of double mutants showed both Als3 and Hyr1 were involved in these processes although Als3 was more important to endocytosis. However Hyr1 mutants were not defective in toxicity or cytokine release for epithelial cells. To further asses the function of Hyr1, assays with neutrophils showed the presence of Hyr1 protects C.. albicans from neutrophil killing independently from neutrophilic c-Met. However, It is not clear why neutrophil c-Met would be expected to be involved in this interaction as it is unknown whether it functions in a similar manner to epithelial c-Met.

Finally in vivo infection experiments are done with hyr1 and als3 mutants. Both single deletion mutants have a modest reductio in CFUs, while the double mutant is not less virulent in terms of CFUs than single mutants. Next, the asl3/hyr1 double mutant CFUs are assessed in neutropenic mice. The rationale for this experiment is not clear since neutropenic mice will still have some neutrophils with activity. The double mutant has very slightly fewer CFU than WT so premature to conclude that the differences in CFUs are a result of differences in phagocyte killing. Similarly mice treated with dual inhibitors of EGFR and c-Met have slightly reduced CFU (is this with WT C. albicans?) that don't correspond with the histopathology sections. Since animals are ingesting systemic drugs, it is not clear how specific the effects are to oral epithelium.

Reviewer #3: In this study the Filler group examine the role of c-Met and Hyr1 in the oropharyngeal candidiasis model in vitro and mice. The findings reveal a novel role of c-Met in endocytosis of Candida albicans which is mediated by Hyr1. The authors perform a series of elegant experiments in vitro, ex vivo, and in vivo that convincingly support their conclusions. The paper is very well written, the figures are of outstanding quality, and the work is novel and important, advancing our understanding of host-fungal interactions at the oral epithelial barrier. I only have very minor comments for the consideration of the authors.

1- In Figure 1F, can the authors show or provide a reference that adding the ligand HGF increases the levels or phosphorylation of c-Met?

2- In Figure 9, the authors nicely show that inhibition of both c-Met and EGFR is required in vivo for a phenotype in infected mice. Since the authors have showed data earlier in their paper on conditional KO mice for c-Met in neutrophils, if the authors still have this colony of mice available, it would be worth infecting them in the OPC model to examine CFUs post-infection after steroids or even in immunocompetent mice. One may expect that these mice won't have a phenotype given the pharmacological inhibition data of Fig 9 but sometimes there are effects that can be revealed with genetic manipulation at the cell specific level that may be masked or missed otherwise. If the mice are not available anymore, the authors could discuss these experiments (alongside conditional KO mice for c-Met in epithelium crossed with Krt-13 or Krt-14 mice) in the discussion.

**Part II – Major Issues: Key Experiments Required for Acceptance**

Reviewer #1: Major points:

Figure 1) it looks like phospho-c-Met peaks at 10 minutes, but the quantification shows an increase over the entire timepoint—is this really a representative blot?

Figure 2) It would be nice to see in Figure 2E whether c-Met, when transfected alone, also is phosphorylated in response to Candida challenge. It is important to know whether c-Met is activatable, not just whether endocytosis occurs.

Figure 4) The authors start by talking about Ssa1, but then it is dropped. It might be worth re-writing to either focus only on the Als3, or add additional data on the role of Ssa1.

The title of Figure 4 is that Hyr1 interacts with c-Met and EGFR, but the data in this figure doesn’t explicitly show Hyr1-EGFR interaction, only Hyr1-cMet.

Figure 5) The model shows that the three fungal proteins are needed to coordinate the complex formation of the three host receptors. The authors should include immunofluorescence experiments, like in Figure 1, showing that the host receptors no longer co-localize when the fungal proteins are deleted. The authors should also show that the co-localization does not occur when one of the host receptors is deleted/depleted.

Figure 8) although the text says there is no difference in CFU at the early time points, the immunofluorescence shows a decrease in anti-Candida staining in the mutant strains. Potentially there is some clearance that is beyond the resolution of the CFU analysis? This should be discussed.

Reviewer #2: Co-Ip experiments were convincing using hyphal Candida cells, however is the same association found with yeast from cells? This specificity is shown for phosphorylation, but is it also true for protein association?

If Hyr1 like Asl3 protects from neutrophil killing and in epithelial cells is involved in endocytosis, it is very likely that this protect in neutrophils is a result of decreased phagocytosis. A phagocytic experiment would add to the understanding of the basis for its function in neutrophils.

The animal experiments are not compelling is associating c-Met or EGFR with infection levels of tested hyphal proteins. As animals are ingesting systemic drugs, it is not clear how specific the effects are to oral epithelium- can this be shown on tongues as done for Fig. 8?

Reviewer #3: (No Response)

**Part III – Minor Issues: Editorial and Data Presentation Modifications**

Reviewer #1: Minor points:

Figure 1) please include uninfected control images. Please also show representative activation of cMET in response to HGF.

Supplemental figure 2B and C are inconsistent between the data, the text, and the significance values shown on the graph. The text says that ‘neither inhibitor reduced C. albicans adherence’ but the authors show a statistically significant difference.

Figure 2I) the bimodal phosphorylation of EGFR in the uninfected cells is potentially interesting, and should be commented on.

Figure 5) typo in the middle—Als3 vs. Als31

Figure 6C) asterisks are cropped

Typo line 202 ‘interacting of with’

Reviewer #2: Fig. 8 A and B, -please add some marking on images indicating where basement membrane is to delineate epithelium from connective tissues to orient reader to where c-Met and EFGR are being expressed.

Reviewer #3: (No Response)

PLOS authors have the option to publish the peer review history of their article (what does this mean?). If published, this will include your full peer review and any attached files.

Reviewer #1: No

Reviewer #2: No

Reviewer #3: No

Figure Files:

Data Requirements:

Reproducibility:

References:

---

## [Editor Report · Decision Letter 1]

25 Jul 2023

Dear Dr. Filler,

We are pleased to inform you that your manuscript 'Candida albicans stimulates formation of a multi-receptor complex that mediates epithelial cell invasion during oropharyngeal infection' has been provisionally accepted for publication in PLOS Pathogens.

Best regards,

Bruce S Klein

Academic Editor

PLOS Pathogens

Alex Andrianopoulos

Section Editor

PLOS Pathogens

Kasturi Haldar

Editor-in-Chief

PLOS Pathogens

orcid.org/0000-0001-5065-158X

Michael Malim

Editor-in-Chief

PLOS Pathogens

orcid.org/0000-0002-7699-2064
---

## [Editor Report · Acceptance letter]

18 Aug 2023

Dear Dr. Filler,

We are delighted to inform you that your manuscript, "Candida albicans stimulates formation of a multi-receptor complex that mediates epithelial cell invasion during oropharyngeal infection," has been formally accepted for publication in PLOS Pathogens.

Best regards,

Kasturi Haldar

Editor-in-Chief

PLOS Pathogens

orcid.org/0000-0001-5065-158X

Michael Malim

Editor-in-Chief

PLOS Pathogens

orcid.org/0000-0002-7699-2064